# Impaired astrocytic Ca²⁺ signaling in awake-behaving Alzheimer's disease transgenic mice

Knut Sindre Åbjørsbråten[1†], Gry HE Syverstad Skaaraas[2†], Céline Cunen[3,4], Daniel M Bjørnstad[1], Kristin M Gullestad Binder[1], Laura Bojarskaite[1,5], Vidar Jensen[1], Lars NG Nilsson[6], Shreyas B Rao[2], Wannan Tang[1,7], Gudmund Horn Hermansen[3], Erlend A Nagelhus[1‡], Ole Petter Ottersen[8], Reidun Torp[2], Rune Enger[1*]

[1]GliaLab at the Letten Centre, Division of Anatomy, Department of Molecular Medicine, Institute of Basic Medical Sciences, University of Oslo, Oslo, Norway; [2]Division of Anatomy, Department of Molecular Medicine, Institute of Basic Medical Sciences, University of Oslo, Oslo, Norway; [3]Statistics and Data Science group, Department of Mathematics, Faculty of Mathematics and Natural Sciences, University of Oslo, Oslo, Norway; [4]Norwegian Computing Center, Oslo, Norway; [5]Department of Neurology, Oslo University Hospital, Oslo, Norway; [6]Department of Pharmacology, University of Oslo and Oslo University Hospital, Oslo, Norway; [7]Department of Clinical and Molecular Medicine, Norwegian University of Science and Technology, Trondheim, Norway; [8]Office of the President, Karolinska Institutet, Stockholm, Sweden

**\*For correspondence:**
rune.enger@medisin.uio.no

†These authors contributed equally to this work

‡Deceased

**Competing interest:** The authors declare that no competing interests exist.

**Abstract** Increased astrocytic Ca²⁺ signaling has been shown in Alzheimer's disease mouse models, but to date no reports have characterized behaviorally induced astrocytic Ca²⁺ signaling in such mice. Here, we employ an event-based algorithm to assess astrocytic Ca²⁺ signals in the neocortex of awake-behaving tg-ArcSwe mice and non-transgenic wildtype littermates while monitoring pupil responses and behavior. We demonstrate an attenuated astrocytic Ca²⁺ response to locomotion and an uncoupling of pupil responses and astrocytic Ca²⁺ signaling in 15-month-old plaque-bearing mice. Using the genetically encoded fluorescent norepinephrine sensor GRAB_NE, we demonstrate a reduced norepinephrine signaling during spontaneous running and startle responses in the transgenic mice, providing a possible mechanistic underpinning of the observed reduced astrocytic Ca²⁺ responses. Our data points to a dysfunction in the norepinephrine–astrocyte Ca²⁺ activity axis, which may account for some of the cognitive deficits observed in Alzheimer's disease.

## Editor's evaluation

This article is of broad interest to readers in the astrocyte and Alzheimer's disease (AD) fields, and it utilizes state-of-the-art techniques to simultaneously record astrocyte calcium and animal behavior. The work provides new insight into astrocyte calcium responses in AD, which has important implications for astrocyte pathophysiology. Overall, the data are of high quality and well analyzed.

**eLife digest** Neurodegenerative conditions such as Parkinson's or Alzheimer's disease are characterized by neurons dying and being damaged. Yet neurons are only one type of brain actors; astrocytes, for example, are star-shaped 'companion' cells that have recently emerged as being able to fine-tune neuronal communication. In particular, they can respond to norepinephrine, a signaling molecule that acts to prepare the brain and body for action. This activation results, for instance, in astrocytes releasing chemicals that can act on neurons.

Certain cognitive symptoms associated with Alzheimer's disease could be due to a lack of norepinephrine. In parallel, studies in anaesthetized mice have shown perturbed astrocyte signaling in a model of the condition. Disrupted norepinephrine-triggered astrocyte signaling could therefore be implicated in the symptoms of the disease. Experiments in awake mice are needed to investigate this link, especially as anesthesia is known to disrupt the activity of astrocytes.

To explore this question, Åbjørsbråten, Skaaraas et al. conducted experiments in naturally behaving mice expressing mutations found in patients with early-onset Alzheimer's disease. These mice develop hallmarks of the disorder. Compared to their healthy counterparts, these animals had reduced astrocyte signaling when running or being startled. Similarly, a fluorescent molecular marker for norepinephrine demonstrated less signaling in the modified mice compared to healthy ones.

Over 55 million individuals currently live with Alzheimer's disease. The results by Åbjørsbråten, Skaaraas et al. suggest that astrocyte–norepinephrine communication may be implicated in the condition, an avenue of research that could potentially lead to developing new treatments.

## Introduction

Since astrocytic $Ca^{2+}$ signals were first discovered in the early 1990s, they have been the object of numerous studies exploring their roles in brain physiology and pathophysiology. Importantly, such signals have been shown to occur in response to a wide array of neurotransmitters and trigger the release of substances that affect neuronal signaling and the vasculature. A growing body of evidence suggests that astrocytic $Ca^{2+}$ signals play important roles in higher brain functions such as memory formation and cortical processing, mediated in part through the neuromodulatory systems (*Adamsky et al., 2018*; *Kol et al., 2020*; *Poskanzer and Yuste, 2016*; *Poskanzer and Yuste, 2011*; *Ye et al., 2020*; *Paukert et al., 2014*).

Astrocytic $Ca^{2+}$ signaling in an Alzheimer's disease (AD) mouse model was first described by *Kuchibhotla et al., 2009*, who found pathological $Ca^{2+}$ waves originating at amyloid plaques, and a general increase in astrocytic $Ca^{2+}$ signaling. Later, it was demonstrated that plaque-associated astrocytic hyperactivity was mediated through activation of metabotropic purine receptors (*Delekate et al., 2014*; *Reichenbach et al., 2018*). These studies were performed under anesthesia, which severely attenuates physiological $Ca^{2+}$ signals (*Thrane et al., 2012*).

The field of astrocytic $Ca^{2+}$ signaling is undergoing a revolution as developments in optical imaging and genetically encoded fluorescent sensors now allow us to monitor these signals in awake-behaving mice, without the confounding effects of anesthesia (*Srinivasan et al., 2015*; *Bojarskaite et al., 2020*). Such studies have revealed exceedingly rich and complex astrocytic $Ca^{2+}$ signaling ranging from large activations of nearly all astrocytes in a field of view (FOV) under locomotion and startle responses due to noradrenergic activity (*Ding et al., 2013*; *Paukert et al., 2014*), to small, localized signals occurring spontaneously or as a response to local neuronal activity (*Bindocci et al., 2017*; *Stobart et al., 2018*; *Srinivasan et al., 2015*). New analytical tools now also enable us to accurately quantify and describe these signals (*Wang et al., 2019*; *Bjørnstad et al., 2021*).

The brain noradrenergic system is crucial for mediating responses to external environmental stimuli, optimizing central nervous system performance and thus for arousal and cognition. Interestingly, norepinephrine is one of the main drivers of astrocytic $Ca^{2+}$ signaling activating astrocytic α1-adrenergic receptors (*Ding et al., 2013*; *Thrane et al., 2012*; *Paukert et al., 2014*). Notably, noradrenergic signaling in relation to locomotion and startle responses causes global increases of astrocytic $Ca^{2+}$ (*Ding et al., 2013*; *Srinivasan et al., 2015*). The main downstream effects of this astrocytic $Ca^{2+}$ activity are not yet well understood, but thought to support neurons metabolically (*O'Donnell et al., 2012*), or through dynamic changes of the size and composition of the extracellular fluids

(*Wang et al., 2012*). Nonetheless, astrocytes are believed to be key actuators of the noradrenergic system (*O'Donnell et al., 2012*; *Wahis and Holt, 2021*). Importantly, NA signaling is assumed to play a pivotal role in AD pathology (*Weinshenker, 2018*; *Peterson and Li, 2018*; *Holland et al., 2021*). Profound loss of noradrenergic neurons is a hallmark of AD that occurs early in the development of the disorder, likely accounting for some of the symptoms of AD, including loss of cognitive function (*Matchett et al., 2021*). However, the mechanisms and signaling pathways connecting perturbed NA and cognition are only rudimentarily understood.

As astrocytes are tightly associated with NA signaling and AD is characterized by perturbed NA system, we set out to investigate whether the norepinephrine–astrocyte $Ca^{2+}$ signaling axis was perturbed in ~15-month-old unanesthetized awake-behaving tg-ArcSwe mice compared to wild-type (WT) littermates. The tg-ArcSwe mice carry two mutations in the amyloid precursor protein gene, the Arctic (E693G) and Swedish (KM670/6701NL) mutations, and exhibit amyloid-β (Aβ) deposits, a hallmark of AD (*Lord et al., 2006*; *Yang et al., 2011*; *Lillehaug et al., 2014*; *Philipson et al., 2009*). The mice have a robust phenotype with emergence of Aβ plaques at 6–7 months of age both in the parenchyma and in the walls of blood vessels, which are biochemically similar to human Aβ plaques (*Philipson et al., 2009*). The prominent cerebral amyloid angiopathy (CAA) in close proximity to astrocytic endfeet makes the model suitable for studying astrocytic signaling (*Yang et al., 2011*).

We demonstrate an attenuated behaviorally induced $Ca^{2+}$ signaling in cortical astrocytes during locomotion. As noradrenergic signaling is known to trigger astrocytic $Ca^{2+}$ signaling in startle responses and locomotion, and pupil responses are regarded a faithful, although indirect, readout of noradrenergic signaling in the brain during physiological conditions (*Reimer et al., 2016*; *Zuend et al., 2020*; *Costa and Rudebeck, 2016*), we compared the pupil responses to the astrocytic $Ca^{2+}$ signals during running and startle and found a positive correlation in WT mice. No such correlation was present in the AD mice. Using the fluorescent extracellular norepinephrine sensor GRAB_NE (*Feng et al., 2019*), we found attenuated norepinephrine signaling in relation to locomotion and startle responses in the AD mice, likely explaining the observed reduced astrocytic $Ca^{2+}$ signaling. Such perturbed behaviorally induced norepinephrine-linked astrocytic $Ca^{2+}$ signaling may account for some of the cognitive deficiencies observed in AD patients.

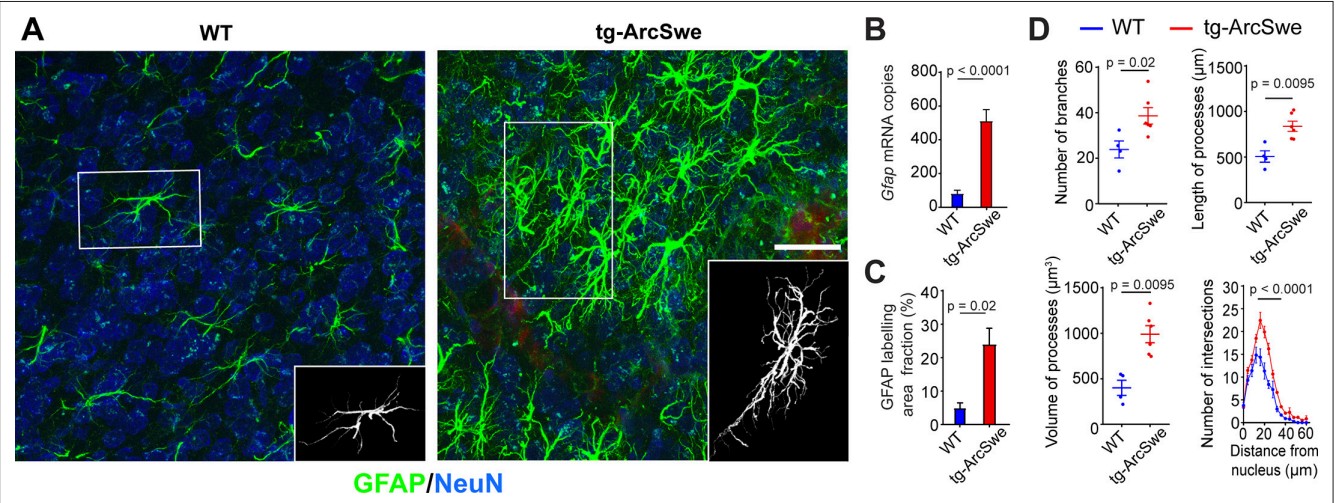

**Figure 1.** Widespread reactive astrogliosis in tg-ArcSwe mice. (**A**) Representative micrographs of 15-month-old wildtype (WT) and tg-ArcSwe mice somatosensory cortex labeled with anti-GFAP antibodies (green) and anti-NeuN antibodies (blue). Scale bar: 40 µm. (**B**) *Gfap* mRNA expression was considerably higher in tg-ArcSwe mice (p<0.0001, n = 8 animals in both groups, Mann–Whitney *U*-test). (**C**) The area fraction of GFAP labeling was significantly higher in tg-ArcSwe mice compared to controls (n = 6 tg-ArcSwe and 4 WT littermates). (**D**) Astrocytes were isolated (inset in **A**) and analyzed with Simple Neurite Tracer (SNT) plugin in Fiji ImageJ. Tg-ArcSwe mice displayed an increase in the total number of branches (p=0.02), the length and volume of processes (p=0.0095 and 0.095, respectively, two-way ANOVA) of the GFAP-labeled astrocytes (n = 8 astrocytes from each animal; six tg-ArcSwe and four WT littermates), as well as an increased number of branching points and intersections at 16–32 µm distance from the nucleus (p<0.0001, two-way ANOVA).

## Results

### Widespread reactive astrogliosis in tg-ArcSwe mice

AD transgenic mouse models with high levels of Aβ deposition exhibit reactive astrogliosis (*Rodríguez-Arellano et al., 2016*). As astrogliosis might affect astrocytic Ca²⁺ activity (*Shigetomi et al., 2019*; *Sano et al., 2019*) and has not been quantified in neocortex of tg-ArcSwe mice yet, we measured glial fibrillary acidic protein (GFAP) expression by immunofluorescence and mRNA levels, and performed morphometric analyses of astrocytes in tg-ArcSwe mice and WT littermates (*Figure 1*).

We found a strong increase in levels of *Gfap* mRNA (81.58 ± 19.56 for WT; 512.13 ± 66.95 for tg-ArcSwe, p<0.0001, n = 8 animals in both groups) in tg-ArcSwe compared to WT littermates (*Figure 1B*). This was supported by a significantly higher GFAP labeling fraction in tg-ArcSwe animals compared to WT littermates (*Figure 1C*, 24.03% ± 4.80% in tg-ArcSwe, n = 6 mice vs. 4.77% ± 1.58% in WT, n = 4 mice, p=0.02). Morphometric analyses displayed a significantly higher number of labeled astrocytic processes (38.63 ± 3.61 in tg-ArcSwe, n = 6 mice; 23.84 ± 3.72 in WT, n = 4 mice; p=0.02), as well as total length (837.9 µm ± 55.9 µm for tg-ArcSwe; 506.0 µm ± 62.5 µm for WT; p=0.0095) and volume of processes (990.9 µm³ ± 93.8 µm³ for tg-ArcSwe; 401.5 µm³ ± 82.9 µm³ for WT; p=0.0095) in tg-ArcSwe mice compared to WT littermates (*Figure 1D*). In addition, astrocytes in the tg-ArcSwe mice exhibited significantly more branching points at the distance 16–32 µm from the nucleus (p<0.0001).

### Two-photon imaging of awake-behaving tg-ArcSwe mice

To characterize astrocytic Ca²⁺ signaling in awake tg-ArcSwe mice and nontransgenic littermates, we employed two-photon microscopy of cortical layer 1–3 astrocytes in the somatosensory cortex expressing GCaMP6f. Recombinant adeno-associated virus (rAAV) was used to deliver the genetic construct, and the *GFAP* promoter was used to target astrocytes (*Figure 2A*). Aβ plaques were visualized in vivo by methoxy-X04 delivered by intraperitoneal injection (*Figure 2A*). Methoxy-X04 enters the brain and specifically stains parenchymal Aβ plaques and cerebrovascular deposits (*Klunk et al., 2002*), and is widely used for in vivo imaging of transgenic mice with Aβ (*Delekate et al., 2014*; *Kuchibhotla et al., 2009*; *Meyer-Luehmann et al., 2008*). Both tg-ArcSwe mice and littermates were injected with methoxy-X04 to rule out any confounding effects of methoxy-X04 on astrocytic Ca²⁺ activity. Imaging was performed at ~30 Hz frame rate to capture fast populations of astrocytic Ca²⁺ transients with simultaneous surveillance video recording of mouse behavior, movement of the treadmill, as well as pupil diameter (*Figure 2B*) to monitor the level of arousal (*Reimer et al., 2016*). The mice were allowed to spontaneously move on a custom-built disc-shaped treadmill, and all mice exhibited both periods of quiet wakefulness (absence of locomotion) and running (*Figure 2—figure supplement 1*). Astrocytic Ca²⁺ signals were analyzed using a newly developed event-based Ca²⁺ signal analysis toolkit, outlining so-called regions of activity (ROAs), combined with manually drawn regions of interest (ROIs) defining astrocytic subcompartments (*Bjørnstad et al., 2021*; *Bojarskaite et al., 2020*). This method detects Ca²⁺ signals in a pixel-by-pixel fashion by (1) estimating the noise level per pixel over time, (2) binarizing the data into signal/non-signal by noise level-based thresholds per pixel, and (3) connecting adjoining active pixels in space and time resulting in *x-y-t* ROAs. For the remainder of this article, we primarily report the *ROA density*, which is the fraction (in %) of the compartment analyzed with signal at any given time. See 'Materials and methods' for more details.

The potential differences in astrocytic Ca²⁺ signaling between the two genotypes during spontaneous runs and startle responses, as well as the relationship between Ca²⁺ signaling and pupil dilation, were investigated through statistical models. These results are presented in the following sections, and the methods are described in detail in 'Statistical analyses'. The Ca²⁺ signal responses were modeled by linear mixed effects regression models. The coefficients of primary interest were the effect of genotype, and the interaction between the effect of pupil dilation and genotype (i.e., the parameter indicating potential differences in the effect of pupil dilation on Ca²⁺ signaling between the two genotypes). In addition, the models included some technical and biological covariates, like the µm per pixel (images acquired with two different levels of magnification), the depth of the measurements, and the maximal speed of mouse locomotion in the time window studied. Further, the models include random intercepts for each mouse. The sensitivity of our result to these modeling choices was assessed by various robustness checks, primarily by iteratively removing individual mice and trials to

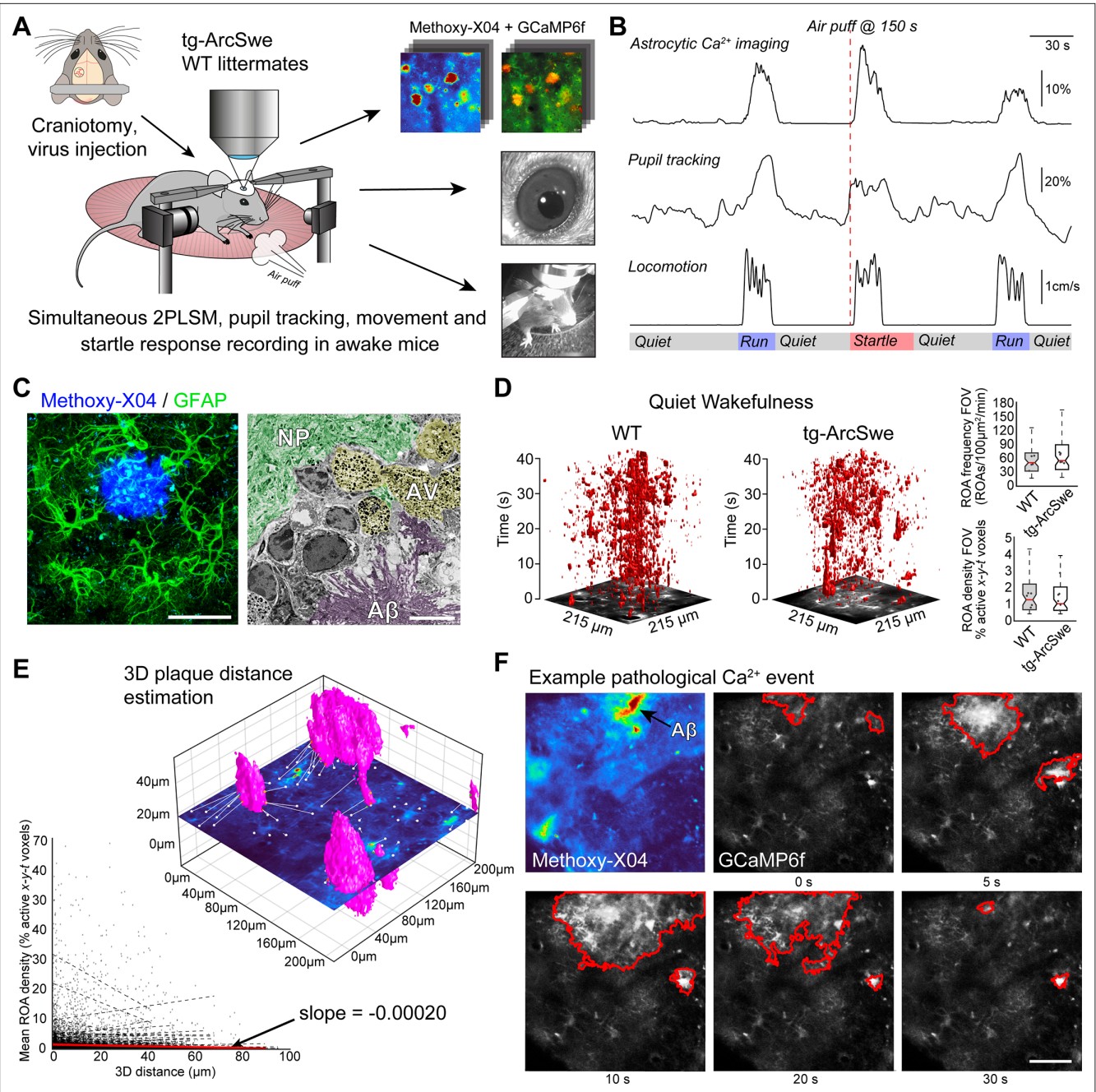

**Figure 2.** Experimental setup and astrocytic Ca²⁺ signaling in quiet wakefulness. (**A**) A craniotomy was performed and virus encoding GCaMP6f was injected into the somatosensory cortex of tg-ArcSwe mice and wildtype (WT) littermates. After 3 weeks of recovery, mice were habituated to head fixation on a disc-shaped treadmill, allowing the mice to move freely at will. Methoxy-X04 was injected 24 hr prior to imaging to visualize amyloid-β (Aβ) plaques. During imaging, both locomotor activity and pupil responses were recorded. At 150 s in 300 s recordings, the mice were subjected to an air puff to induce a startle response. (**B**) A representative imaging trial. Simultaneous recording of astrocytic Ca²⁺ signaling, pupil diameter, and mouse locomotion during quiet wakefulness, spontaneous locomotion, and air puff-induced startle. (**C**) The mice were ~15 months of age during experiments, at a time when they exhibited dense-core Aβ plaques. Left image: confocal micrograph of an Aβ plaque (methoxy-X04, blue) and astrocytes (anti-GFAP, green). Scale bar: 40 μm. Right image: electron micrograph showing a dense Aβ plaque (purple overlay), autophagic vacuoles (yellow overlay), and relatively normal neuropil morphology (green overlay). Scale bar: 2 μm. (**D**) Astrocytic Ca²⁺ signals during quiet wakefulness (absence of locomotion) in the form of regions of activity (ROAs) displayed in an *x-y-t* 3D rendering where red regions denote signal. Box-and-whisker plots representing overall Ca²⁺ signals in quiet wakefulness in tg-ArcSwe mice and littermates. (**E**) 3D visualization of the imaging plane relative to Aβ plaques, with lines representing shortest distance from plaque to region of activity (ROI). We found a low correlation between distance to nearest plaque and the overall level of astrocytic Ca²⁺ signaling in quiet wakefulness. (**F**) Example of pathological astrocytic Ca²⁺ wave emanating from an Aβ plaque. Methoxy-X04

*Figure 2 continued on next page*

*Figure 2 continued*

visualized by 2PL (upper left) and time series from the same field of view (FOV) of astrocytic Ca²⁺ imaging with a pathological signaling event outlined in red. Scale bar: 50 µm.

The online version of this article includes the following source data and figure supplement(s) for figure 2:

**Source data 1.** A .csv file containing Ca²⁺ response data (region of activity [ROA] density and frequency), locomotion data, and covariates used for statistical modeling for quiet wakefulness.

**Figure supplement 1.** Locomotor behavior in wildtype (WT) and tg-ArcSwe mice.

**Figure supplement 2.** Robustness and sensitivity analyses for the uncoupling between pupil dilation and the region of activity (ROA) density rise rate.

**Figure supplement 3.** GCaMP6f expression level assessed by GFP labeling.

**Figure supplement 4.** Regions of activity (ROAs) during quiet wakefulness exhibit the same characteristics in both genotypes.

see to which degree this influenced p-values and estimates (see 'Statistical analyses' and *Figure 2— figure supplement 2*).

## Astrocytes close to Aβ plaques express GCaMP6f

The tg-ArcSwe mice were imaged at ~15 months of age. At this age, they present with Aβ plaques throughout the cortical mantle, and score poorly on behavioral tasks (*Codita et al., 2010*; *Lillehaug et al., 2014*; *Lord et al., 2006*; *Figure 2C*). Aβ plaques were characterized by loss of cells and severely perturbed tissue morphology, including autophagic vacuoles (*Figure 2C*). Even so, relatively normal cellular morphology was present at short distances away from Aβ plaques, and astrocytes faithfully expressed the GCaMP6f Ca²⁺ sensor 3 weeks after viral transduction (*Figure 2A and C*, *Figure 2— figure supplement 3*). Of note, GCaMP6f expression as judged by average GFP labeling and area fraction of GFP labeling in confocal micrographs were similar in the two groups (*Figure 2—figure supplement 3*).

## Astrocytic Ca²⁺ signals in quiet wakefulness are preserved in tg-ArcSwe mice

In quiet wakefulness (defined as absence of locomotion), the gross level of astrocytic Ca²⁺ signaling was similar in mutant mice and their littermates as measured by ROA frequency, ROA density (the active fraction of a compartment), as well as event size and duration in the full FOV and across the different astrocytic subcompartments (*Figure 2D*, *Figure 2—figure supplement 4*). We were neither able to detect a clear correlation in overall astrocytic Ca²⁺ signaling measured by ROA density and the distance from nearest Aβ plaque (in 3D) (slope = –0.00020, *Figure 2E*). However, we found some examples of long-lasting pathological Ca²⁺ waves as reported previously in anesthetized mice (*Delekate et al., 2014*; *Kuchibhotla et al., 2009*). Such Ca²⁺ waves were found in 10–15% of recordings from tg-ArcSwe mice (*Figure 2F*), and the number of such clear pathological events was very low compared to the overall astrocytic Ca²⁺ signaling.

## Uncoupling between pupil dilation and astrocytic Ca²⁺ responses during spontaneous running in tg-ArcSwe mice

Locomotor behavior is known to be strongly correlated with astrocytic Ca²⁺ signaling (*Paukert et al., 2014*; *Bojarskaite et al., 2020*; *Srinivasan et al., 2015*), putatively through the activation of the noradrenergic and cholinergic neuromodulatory systems in conjunction with local network activity (*Kjaerby et al., 2017*). To investigate whether the physiological astrocytic Ca²⁺ responses were preserved in the tg-ArcSwe mice, they were allowed to move freely on a custom built disc-shaped treadmill (*Bojarskaite et al., 2020*). All mice exhibited both running and behavioral quiescence, and the level of running between the two genotypes was comparable (*Figure 2—figure supplement 1*). Running was accompanied by an increase in pupil size and a brisk increase in astrocytic Ca²⁺ signaling typically involving most of the astrocytes in the FOV in both genotypes (*Figure 3A,B*). Overall, astrocytic Ca²⁺ responses to spontaneous locomotion were somewhat reduced in tg-ArcSwe mice compared to WT littermates: When astrocytic Ca²⁺ signals were analyzed using a linear mixed effects regression model (*Table 1*), a lower ROA density rise rate was found in tg-ArcSwe mice when assessing the full FOV (0.31 in WT vs. 0.20 in tg-ArcSwe, p=0.032), and astrocytic processes (0.32 in WT vs. 0.20 in tg-ArcSwe,

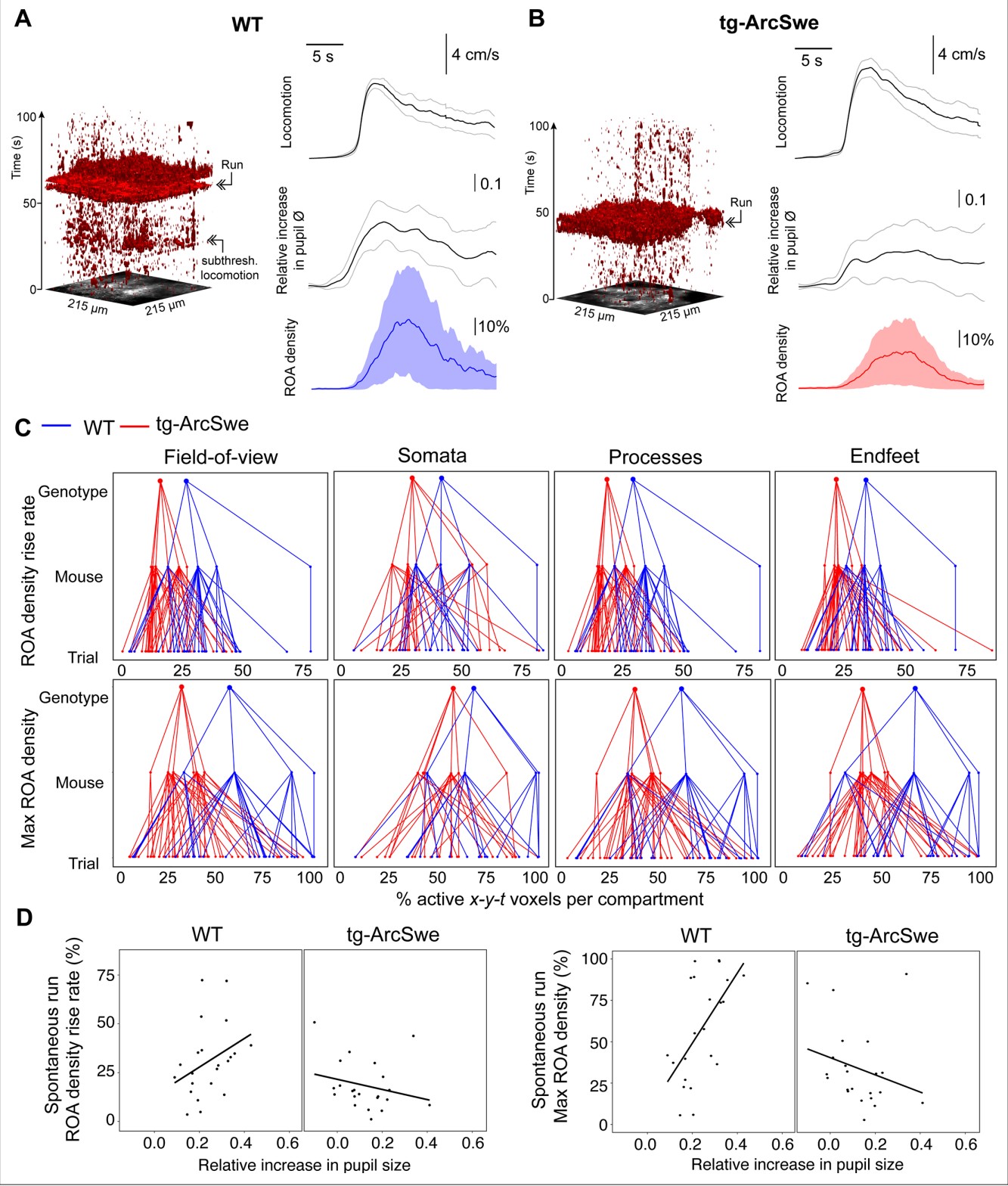

**Figure 3.** Uncoupling of pupil dilation and astrocytic Ca²⁺ responses during spontaneous running. (**A**) Left: Representative *x-y-t* rendering of regions of activity (ROAs) during quiet wakefulness and spontaneous locomotion in wildtype (WT) littermates and tg-ArcSwe mice, where red regions denote signal. A spontaneous run occurred at about 60 s. The slight increase in ROA density at 25 s is due to small locomotor activity that was not large enough to be defined as a run. Right: mean time course of (top-to-bottom) locomotion expressed as wheel speed, relative increase in pupil diameter, and astrocytic Ca²⁺ signals expressed as ROA density. The locomotion trace shows mean response across all trials with 95% confidence interval, pupil trace,

*Figure 3 continued on next page*

*Figure 3 continued*

and ROA density traces show median across all trials ± median absolute deviation. (**B**) Same as (**A**), but for tg-ArcSwe mice. A spontaneous run occurred at about 40 s. (**C**) Hierarchical plots showing median value per genotype in upper level, median value per mouse in middle level, and median level per trial in lower level, in the two genotypes. Upper row: plots showing the median levels of ROA density rise rate in (left-to-right) the full FOV, astrocytic somata, astrocytic processes, and astrocytic endfeet. Lower row: Same as upper row, but showing median levels of max ROA density. (**D**) Scatterplots of ROA density rise rate (left) and max ROA density (right) vs. relative increase in pupil size upon spontaneous running. See *Table 2* for statistical analyses and p-values.

The online version of this article includes the following source data and figure supplement(s) for figure 3:

**Source data 1.** A .csv file containing Ca$^{2+}$ response data (region of activity [ROA] density and frequency), locomotion data, and covariates used for statistical modeling for spontaneous running.

**Figure supplement 1.** Mean region of activity (ROA) activation in active wakefulness.

p=0.032), whereas the Ca$^{2+}$ responses were not significantly different in astrocytic somata and endfeet (0.46 in WT vs. 0.39 in tg-ArcSwe, p=0.23, and 0.35 in WT vs. 0.30 in tg-ArcSwe, p=0.11, respectively) (*Figure 3C*). Max ROA density values in WT vs. tg-ArcSwe were significantly different when assessing the full FOV (0.63 vs. 0.42, p=0.033), near significantly different when assessing astrocytic processes and endfeet (0.72 vs. 0.53, p=0.053 for processes and 0.69 vs. 0.51, p=0.068 for endfeet), and not significantly different for astrocytic somata (0.81 vs. 0.71, p=0.25) (*Figure 3C*). For mean ROA density values, see *Figure 3—figure supplement 1*.

Pupil responses are known to be a faithful indirect indicator of activity in the locus coeruleus in mice, even though also the cholinergic neuromodulatory system plays a role for sustained pupil dilation (*Reimer et al., 2016*). As norepinephrine is known to be a potent trigger of astrocytic Ca$^{2+}$ signaling (*Bekar et al., 2008*; *Srinivasan et al., 2015*; *Paukert et al., 2014*), one would expect to find a correlation between pupil dilations and astrocytic Ca$^{2+}$ signals. This was indeed the case for WT mice (*Table 2*): When comparing ROA density rise rate and pupil dilation, we found a positive slope in line with previous reports (*Zuend et al., 2020*; *Figure 3D*). In transgenic mice, this correlation was lost, or even reversed (*Figure 3D*, *Table 2*: slope of 0.80 in WT vs. –0.35 in tg-ArcSwe, p=0.007), demonstrating an uncoupling between pupil responses and astrocytic Ca$^{2+}$ signaling in tg-ArcSwe mice. Similarly, when assessing max ROA density, we found a positive slope in WT, which was lost in tg-ArcSwe (*Figure 3D*, 2.06 in WT vs. –0.58 in tg-ArcSwe, p=0.00039). Similar

**Table 1.** Estimated statistical model for region of activity (ROA) density rise rate in the field of view (FOV) during spontaneous running.
μm per pixel has a small/no effect, but we include it for consistency. Deeper measurements tend to have lower ROA density rise rate. Given that the mice are running, a higher speed does not appear to be associated with a higher ROA density rise rate (n = 109, in 10 mice).

| | Estimate | p-Value |
|---|---|---|
| Intercept (TG) | 0.1969 | |
| Genotype (WT) | 0.1085 | 0.0318 |
| μm per pixel | –0.0055 | >0.10 |
| Depth | –0.0432 | 0.0002 |
| Maximum speed | 0.0000 | >0.10 |

**Table 2.** Estimated statistical model for region of activity (ROA) density rise rate in the field of view (FOV) and pupil dilation during spontaneous running.
When including pupil dynamics in the model (cf. *Table 1* without pupil dynamics), the effect of genotype was no longer significant. However, the relationship between pupil dilation and ROA density rise rate was different in the two genotypes, with the wildtype (WT) mice having a significantly higher slope. μm per pixel and depth of the recording had no significant effect on the model, but we included them for consistency. The slope for the WT group is found by adding the slope in the TG group with the interaction term (e.g., –0.3537 + 1.1562 = 0.80) (n = 44, in 10 mice).

| | Estimate | p-Value |
|---|---|---|
| Intercept (TG) | 0.2297 | |
| Genotype (WT) | –0.1245 | >0.10 |
| Pupil dilation (TG) | –0.3537 | >0.10 |
| Pupil dilation × genotype (WT) | 1.1562 | 0.0070 |
| μm per pixel (binary) | 0.0306 | >0.10 |
| Depth | –0.0409 | 0.0996 |
| Maximum speed | –0.0011 | 0.0351 |

slopes were found when comparing mean ROA density vs. pupil dilation (*Figure 3—figure supplement 1C*).

## Uncoupling between pupil dilation and astrocytic Ca²⁺ responses during startle in tg-ArcSwe mice

Another main trigger for astrocytic Ca²⁺ signals are startle responses that also are mediated through an activation of the noradrenergic system (*Ding et al., 2013*; *Srinivasan et al., 2015*). Even though typically triggering running, the startle response could also trigger freezing behavior and is thought to activate different subcortical networks than spontaneous locomotor behavior (*Caggiano et al., 2018*; *Ferreira-Pinto et al., 2018*; *Grillner and El Manira, 2020*). To investigate whether the startle-evoked astrocytic Ca²⁺ responses were preserved in the tg-ArcSwe mice, they were subjected to 10 air puffs delivered at 10 Hz directed to the vibrissa, nasal, and facial region contralaterally to the recording side once per trial at 150 s in a 300 s two-photon imaging recording. Trials in which the mouse was spontaneously running at or immediately before the air puff were excluded from the analyses. We found no signs of habituation to the stimulus in terms of behavioral response (*Figure 4—figure supplement 1*). Interestingly, tg-ArcSwe mice were more prone to start running during startle responses than WT littermates (*Figure 2—figure supplement 1*), consistent with previous reports of enhanced startle response in other mouse models of AD (*McCool et al., 2003*). The level of pupil dilation was however similar in the two genotypes (0.17 vs. 0.12 relative increase in pupil size in WT vs. tg-ArcSwe, respectively, p=0.36, 86 trials) (*Figure 4A and B*). Regarding astrocyte Ca²⁺ response to startle, we found no differences between the genotypes. When modeled with a mixed effects linear regression model (*Table 3*), for ROA density rise rate, Ca²⁺ responses in the full FOV and in all astrocytic subcompartments were statistically similar between the genotypes (FOV: 0.34 in WT vs. 0.23 in tg-ArcSwe, p=0.09; somata: 0.25 in WT vs. 0.23 in tg-ArcSwe, p=0.8; processes: 0.42 in WT vs. 0.31, p=0.09; endfeet: 0.36 in WT vs. 0.25, p=0.10) (*Figure 4C*). Similarly, max ROA density was similar for the full FOV (0.36 in WT vs. 0.35 in tg-ArcSwe, p=0.40), astrocytic somata (0.37 in WT vs. 0.35 in tg-ArcSwe, p=0.83), astrocytic processes (0.41 in WT vs. 0.34 in tg-ArcSwe, p=0.39), and astrocytic endfeet (0.41 in WT vs. 0.35 in tg-ArcSwe, p=0.48) (*Figure 4C*). For mean ROA density values, see *Figure 3—figure supplement 1*.

However, the relationship between astrocytic Ca²⁺ responses and pupillary responses were different in the two genotypes (*Figure 4D*). WT mice displayed a clear positive slope, while tg-ArcSwe exhibited a slope close to zero (0.91 vs. -0.013 p = 0.00043 for ROA density rise rate, and 1.27 vs. 0.083 p = 0.0043 and max ROA density, respectively), suggesting a potential uncoupling between pupillary and astrocytic Ca²⁺ responses during startle in tg-ArcSwe mice, similar to what was observed during spontaneous running (*Figure 3D*). Similar slopes were found when comparing mean ROA density vs. pupil dilation (*Figure 3—figure supplement 1*).

## Attenuated noradrenergic signaling in tg-ArcSwe mice

To investigate the cause of our observed attenuated astrocytic Ca²⁺ responses during locomotion and startle responses, and uncoupling of pupil responses and astrocytic Ca²⁺ responses, we took advantage of the newly developed genetically encoded fluorescent extracellular norepinephrine sensor, the GRAB$_{NE}$ sensor (*Feng et al., 2019*). The sensor was delivered by rAAV transduction of the GRAB$_{NE}$ (version 1m) construct and targeted to membranes of neurons using the *hSyn1* promoter. The mice were subjected to the same imaging protocols as for Ca²⁺ imaging, and we recorded norepinephrine fluorescence, pupil dynamics, and locomotor activity. Differences between the genotypes were modeled by linear mixed effects regression models as outlined for Ca²⁺ imaging (results presented in *Tables 5 and 6*). The coefficients modeled were the effect of genotype, and the interaction between the effect of pupil dilation and genotype (i.e., the parameter indicating potential differences in the effect of pupil dilation on norepinephrine signaling between the two genotypes). During both spontaneous running and startle responses, we observed increases in norepinephrine fluorescence in WT mice (*Figure 5*). This effect was less prominent in tg-ArcSwe mice, and similar to Ca²⁺ imaging, we found a decoupling between pupil responses and noradrenergic signaling. In WT mice, for spontaneous runs, the relative increase in pupil diameter explained a large proportion of change in norepinephrine signaling (*Table 5*). In tg-ArcSwe mice, pupil responses and norepinephrine signaling were

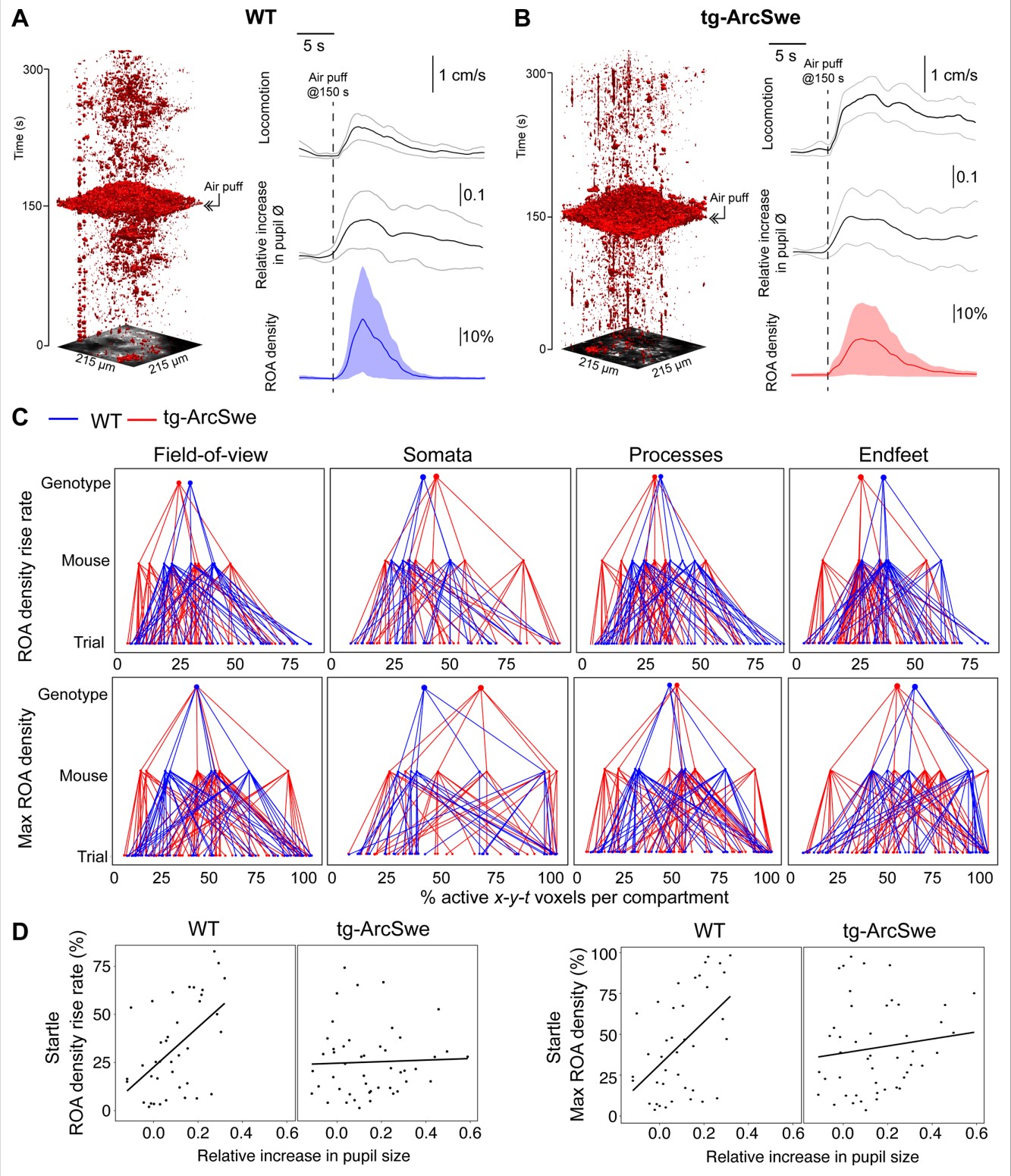

**Figure 4.** Uncoupling of pupil dilation and astrocytic Ca²⁺ responses during startle. (**A**) In wildtype (WT) mice, in an imaging trial of 300 s duration, mice were subjected to an air puff to the face/vibrissa contralateral to the imaging window at 150 s. This caused an increase in locomotor activity, pupil dilation, and a pronounced increase in astrocytic Ca²⁺ signaling. Region of activity (ROAs) are presented as an *x-y-t* 3D rendering where red regions denote signal. Locomotion trace shows mean response across trials with 95% confidence interval, pupil trace, and ROA density trace show median activation across trials ± median absolute deviation. (**B**) Same as (**A**), but for tg-ArcSwe mice. (**C**) Hierarchical plots showing median value per genotype in upper level, median value per mouse in middle level, and median level per trial in lower level, in the two genotypes. Upper row: plots showing the

*Figure 4 continued on next page*

*Figure 4 continued*

median level of the ROA density rise rate in (left-to-right) the full field of view (FOV), astrocytic somata, astrocytic processes, and astrocytic endfeet. Lower row: same as upper row but for max ROA density. (**D**) Scatterplots of ROA density rise rate (left) and max ROA density (right) vs. relative increase in pupil size. See *Tables 3 and 4* for statistical analyses and p-values.

The online version of this article includes the following source data and figure supplement(s) for figure 4:

**Source data 1.** A .csv file containing Ca$^{2+}$ response data (region of activity [ROA] density and frequency), locomotion data, and covariates used for statistical modeling for startle responses.

**Figure supplement 1.** Behavioral response to air puff across trials.

**Figure supplement 2.** The dynamic range of pupil dilation and constrictions in the two genotypes.

**Figure supplement 3.** Method for assessing region of activity (ROA) density and pupil dynamics.

less correlated. The results for startle responses were similar (*Table 6*). These findings suggest that the reduced Ca$^{2+}$ signaling and uncoupling between pupil responses and astrocytic Ca$^{2+}$ signaling in tg-ArcSwe mice could be due to attenuated noradrenergic release in cortex and lower correlation between pupil responses and noradrenergic signaling.

## Discussion

Astrocytic Ca$^{2+}$ signaling is emerging as a key component of signal processing in the brain and figures prominently in brain state transitions and memory formation (*Poskanzer and Yuste, 2016*; *Poskanzer and Yuste, 2011*; *Adamsky et al., 2018*; *Bojarskaite et al., 2020*; *Vaidyanathan et al., 2021*). Aberrant astrocytic Ca$^{2+}$ signaling could hence be implicated in the perturbed cognition seen in dementia. Indeed, previous studies have shown increased astrocytic Ca$^{2+}$ signaling and spreading pathological Ca$^{2+}$ waves in AD mouse models (*Kuchibhotla et al., 2009*; *Delekate et al., 2014*; *Reichenbach et al., 2018*). Based on these studies in anesthetized animals and other studies (*Haughey and Mattson, 2003*; *Lim et al., 2013*; *Abramov et al., 2003*; *Abramov et al., 2004*; *Verkhratsky, 2019*), the current concept is that aberrant signaling to some degree is spatially coupled to amyloid deposits – the pathological hallmark of AD.

Methodological advances now allow astrocytic Ca$^{2+}$ signals to be studied in awake animals. Benefitting from this opportunity, we show that tg-ArcSwe mice sustain a pattern of behaviorally induced astrocytic Ca$^{2+}$ signaling similar to that found in littermate controls. However, the signals are weaker than in controls and do not display the correlation with pupil responses typically seen in WT animals. The behaviorally induced Ca$^{2+}$ signals

**Table 3.** Estimated statistical model for region of activity (ROA) density rise rate in the field of view (FOV) during startle.
Depth from the cortical surface and maximal speed during the startle response had a significant effect in the model, whereas the µm per pixel (level of magnification of the image recording) had no significant effect (n = 117 trials, in 11 mice).

| | Estimate | p-Value |
|---|---|---|
| Intercept (TG) | 0.2295 | |
| Genotype (WT) | 0.1110 | 0.0933 |
| µm per pixel (binary) | –0.0034 | >0.10 |
| Depth | –0.0696 | 0.0004 |
| Maximum speed | 0.0009 | 0.022 |

**Table 4.** Estimated statistical model for region of activity (ROA) density rise rate in the field of view (FOV) and pupil dilation during startle.
When including the pupil dilation in the statistical model, the two genotypes are clearly different. Moreover, depth from the cortical surface had a significant effect in the model, whereas optical zoom and maximum speed had a nonsignificant effect (n = 95 trials, in 11 mice). The slope for the wildtype (WT) group is found by adding the slope in the TG group with the interaction term (e.g., –0.0129 + 0.9235 = 0.91).

| | Estimate | p-Value |
|---|---|---|
| Intercept (TG) | 0.2199 | |
| Genotype (WT) | –0.0279 | >0.1 |
| Pupil dilation (TG) | –0.0129 | >0.1 |
| Pupil dilation × genotype (WT) | 0.9235 | 0.00043 |
| Optical zoom | 0.0036 | >0.1 |
| Depth | –0.0679 | 0.00063 |
| Maximum speed | 0.0006 | 0.0587 |

**Table 5.** Estimated statistical model for change in GRAB<sub>NE</sub> fluorescence and pupil dilation during spontaneous runs.

The coupling between pupil responses and noradrenergic signaling is weaker in tg-ArcSwe mice compared to wildtype (WT) littermates (p=0.043, n = 100 running episodes, 43 trials, in 7 mice). The slope for the WT group is found by adding the slope in the TG group with the interaction term (e.g., 0.0030 + 0.043 = 0.046).

|  | Estimate | p-Value |
|---|---|---|
| Intercept (TG) | 0.0030 |  |
| Genotype (WT) | –0.0012 | >0.1 |
| Pupil dilation (TG) | 0.00022 | >0.1 |
| Pupil dilation × genotype (WT) | 0.043 | 0.042 |
| Increase in speed | –0.00000401 | >0.1 |

**Table 6.** Estimated statistical model for change in GRAB<sub>NE</sub> fluorescence and pupil dilation during startle.

The coupling between pupil responses and noradrenergic signaling during startle responses is weaker in tg-ArcSwe mice compared to wildtype (WT) littermates (p=0.037, n = 59 trials, in 7 mice). The slope for the WT group is found by adding the slope in the TG group with the interaction term (e.g., 0.0078 + 0.037 = 0.045).

|  | Estimate | p-Value |
|---|---|---|
| Intercept (TG) | 0.0078 |  |
| Genotype (WT) | –0.0063 | >0.1 |
| Pupil dilation (TG) | –0.013 | >0.1 |
| Pupil dilation × genotype (WT) | 0.037 | 0.036 |
| Increase in speed | –0.0000029 | >0.1 |

bear no clear spatial correlation to Aβ plaques. We conclude that elimination of anesthesia unveils a new dimension of Ca²⁺ signaling, super-imposed on the locally induced signals described in previous studies. The uniform attenuation of the behaviorally induced signals in tg-ArcSwe mice and their uncoupling from pupil responses point to a potential perturbation of neuromodulatory control.

Astrocytic Ca²⁺ signaling in awake-behaving mice is dominated by norepinephrine-induced astrocytic Ca²⁺ signals across the cortical mantle in relation to locomotor or startle responses (*Srinivasan et al., 2015*; *Ding et al., 2013*). The downstream effects of these synchronized, global Ca²⁺ signals coupled to arousal are not fully understood, but they may play a role in altering the levels of extracellular K⁺ (*Wang et al., 2012*), release of gliotransmitters such as glutamate or ATP (*Kjaerby et al., 2017*; *Haydon and Nedergaard, 2015*), or metabolic supply (*Zuend et al., 2020*). Independent of what the exact downstream mechanisms are, there are reasons to believe that astrocytes serve as actuators of the noradrenergic neuromodulatory system, presumably exerting noradrenergic effects on neural network processing and ultimately affecting cognitive function (*Ye et al., 2020*; *Holland et al., 2021*; *Poskanzer and Yuste, 2016*).

The perturbed astrocytic Ca²⁺ response during locomotion in AD mice seems to be at least partially explained by a blunted norepinephrine increase upon startle and locomotion as evaluated by GRAB<sub>NE</sub> fluorescence. The primary norepinephrine nucleus in the brain, locus coeruleus, is known to be affected early in AD, both in humans and in animal models (*Weinshenker, 2018*; *Jacobs et al., 2021*; *Braak and Del Tredici, 2011*), and perturbed noradrenergic function is likely an important factor both in disease progression in AD patients and accounting for the cognitive decline of AD patients (*Weinshenker, 2018*; *Peterson and Li, 2018*; *Holland et al., 2021*). Restoring noradrenergic signaling in AD mice with perturbed noradrenergic signaling pharmacologically or by pharmacogenetics has been demonstrated to be beneficial (*Holland et al., 2021*). In our study, we find a largely retained level of pupil response in the AD mice during startle responses and a similar dynamic range of pupil responses (*Figure 4—figure supplement 2*), which under physiological circumstances would suggest a functional noradrenergic system, as a strong correlation between locus coeruleus activity and pupil responses has been established (*Reimer et al., 2016*; *Costa and Rudebeck, 2016*). The underlying connectivity that enables the pupils to be faithful readouts of the noradrenergic system is to the best of our knowledge not fully established, even though spinal projections from locus coeruleus have been demonstrated (*Hancock and Fougerousse, 1976*; *Costa and Rudebeck, 2016*; *Liu et al., 2017*). Our finding of a decoupling between norepinephrine-mediated astrocytic Ca²⁺ signaling and arousal-induced pupillary dilation could be due to perturbed connectivity between locus coeruleus and the relevant nuclei

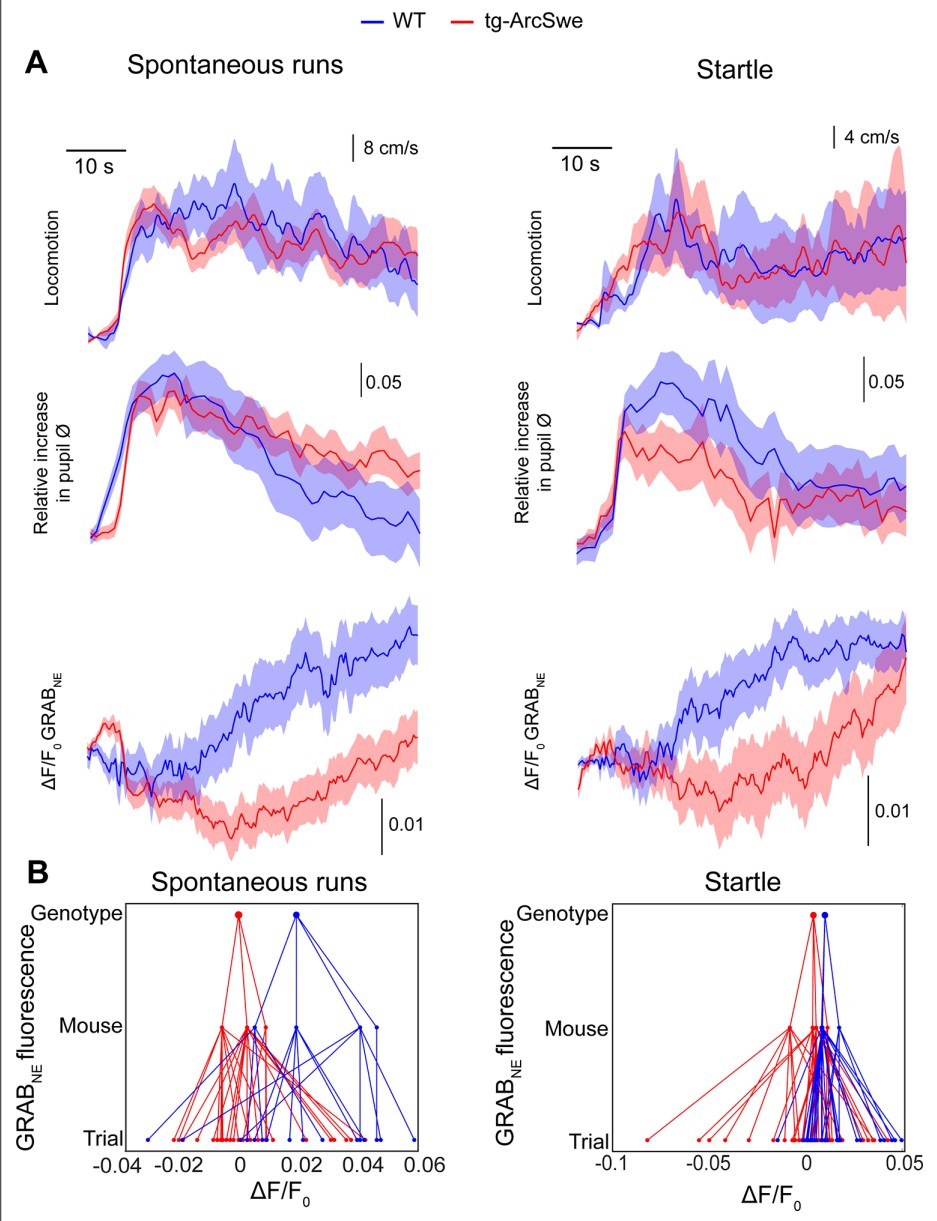

**Figure 5.** Attenuated norepinephrine responses in tg-ArcSwe mice. Mice expressing the fluorescent extracellular norepinephrine sensor GRAB$_{NE}$ were subjected to two-photon imaging trials similar to the astrocytic Ca$^{2+}$ imaging experiments. (**A**) Mean traces ± standard error of locomotor responses, pupil responses and GRAB$_{NE}$ fluorescence during spontaneous runs (left column) and during startle (right column). (**B**) Hierarchical plots showing median ΔF/F$_0$ of GRAB$_{NE}$ fluorescence per (from bottom to top) trial, mouse and genotype, in spontaneous runs and startle responses.

The online version of this article includes the following source data for figure 5:

**Source data 1.** A .mat file containing GRAB$_{NE}$ fluorescence traces during spontaneous running and startle as well as locomotion data and covariates used for statistical modeling.

or projections controlling activity in the superior cervical ganglion and consequently the sympathetic projections to the pupils.

Given the clear differences in norepinephrine signaling between the two genotypes, one would have expected even more pronounced differences in astrocytic Ca$^{2+}$ signaling. This discrepancy could possibly be explained by either the lack of sensitivity of the GRAB$_{NE}$ sensor in the current experimental paradigm to pick up small increases in cortical norepinephrine or potentially by a compensatory

upregulation of adrenergic receptors or other receptors in reactive astrocytes in the tg-ArcSwe mice. In line with this, Reichenbach et al. demonstrated an age/disease stage-related upregulation of P2R1-receptors in an AD mouse model that correlated with the level of reactive astrogliosis (*Reichenbach et al., 2018*). Moreover, potentially the expression of α1-adrenergic receptors in astrocytes is changed in AD, similar to reports of how α2 and β2 receptors are upregulated in microvessels and neurons in AD patients, respectively (*Kalaria et al., 1989*; *Kalaria and Harik, 1989*). Future studies probing the expression levels and patterns of adrenergic receptors and downstream signaling pathways in astrocytes in AD models are warranted.

Although still sparsely investigated, both attenuated and increased $Ca^{2+}$ signaling have been demonstrated in various models of reactive astrogliosis, but so far without a clear understanding of the mechanisms involved (*Shigetomi et al., 2019*). Widespread reactive astrogliosis is nonetheless highly likely to perturb the physiological signaling of astrocytes as prominent changes in the expression of key receptors and molecules of intracellular pathways are known to occur (*Habib et al., 2020*; *Escartin et al., 2021*). In this study, we employed 15-month-old tg-ArcSwe mice and WT littermates. These mice exhibit a relatively high plaque burden and represent a relatively advanced stage of AD. It is possible that the attenuated astrocytic $Ca^{2+}$ responses we observe is a feature of advanced disease, and not earlier stages of the disease development. Similarly, the sparsity of pathological 'hyperactive' astrocytic $Ca^{2+}$ signaling events in our data compared to previous studies (*Delekate et al., 2014*; *Kuchibhotla et al., 2009*; *Reichenbach et al., 2018*) could stem from the other studies being performed at ages that represents earlier stages in the disease progression. Further studies across AD models and stages of disease progression mice are warranted to conclude.

Previous studies have shown mixed results regarding an effect of plaque–astrocyte distance, with reports of both hyperactivity close to plaques (*Delekate et al., 2014*) and no such correlation except for plaques serving as initiation sites for pathological intercellular $Ca^{2+}$ waves (*Kuchibhotla et al., 2009*). In our awake-behaving mice, we found no clear correlation between overall $Ca^{2+}$ activity and 3D distance reconstruction of plaque positions relative to the imaging plane (*Figure 2E*). The apparent discrepancies in the literature and this study may be due to different AD mouse models, different age groups investigated, different $Ca^{2+}$ indicators employed, or lastly, the effects of removing anesthesia allowing for a much richer repertoire of behaviorally induced astrocytic $Ca^{2+}$ signaling to emerge, effectively masking a potential weak correlation.

Astrocytes have a highly complex and specialized morphology, and this morphology is known to change in reactive astrogliosis (*Escartin et al., 2021*). The majority of astrocytic processes are much smaller than what can be clearly delineated by non-super-resolution optical microscopy, but to what extent these small processes are altered in reactive astrogliosis is unknown, although the astrocytic territories are known to be preserved (*Wilhelmsson et al., 2006*). Moreover, GFAP, which was used as the promoter for the GCaMP6f expression, was highly upregulated in tg-ArcSwe mice. This difference in GFAP expression did not translate into obvious differences in sensor expression levels as judged by GFP labeling of brain slices (*Figure 2—figure supplement 3*). We cannot rule out that gliosis-induced morphological changes in combination with potential subtle differences in GCaMP6f expression and potentially an elevated baseline intracellular $Ca^{2+}$ concentration (*Kuchibhotla et al., 2009*) may influence our results. However, the ability for the whole FOV to be activated during spontaneous running and startle responses and the lack of correlation (if anything a negative correlation) between distance to the nearest Aβ plaque and gross level of $Ca^{2+}$ signaling suggest that our findings are not due to degree or pattern of sensor expression.

This study underscores the importance of studying astrocytic $Ca^{2+}$ signals in unanesthetized mice and to carefully consider animal behavior when interpreting astrocytic $Ca^{2+}$ dynamics. By lifting the confounding effects of anesthesia, we found that the astrocytic hyperactivity previously reported in AD mouse models was only a part of the total picture. At first glance, the physiological $Ca^{2+}$ responses were remarkably well preserved and not characterized by a general increase in astrocytic $Ca^{2+}$ signaling. However, behavior like quiet wakefulness and locomotion are not static entities, and the degree of activation of all relevant parameters needs to be taken into account with statistical modeling to be able to conclude if there are relevant differences between the genotypes. We were able to demonstrate attenuated $Ca^{2+}$ dynamics and norepinephrine signaling in the tg-ArcSwe mice, and given the growing spectrum of roles ascribed to the norepinephrine–astrocyte $Ca^{2+}$ signaling

in higher brain functions, the present findings may highlight one potential cause for the cognitive decline of AD patients.

# Materials and methods

## Animals

Tg-ArcSwe mice carry a human *APP* cDNA with the Arctic (p. E693G) and Swedish (p. KM670/671NL) mutations where the human AβPP gene is inherited only from male mice to ensure a more uniform onset of Aβ deposition (*Lillehaug et al., 2014*). The transgenic animals develop parenchymal Aβ plaques from 6 months of age and CAA from 8 months of age (*Yang et al., 2011*; *Lord et al., 2006*). Six tg-ArcSwe mice and five WT littermates (both males and females) were used in this study. Genotyping was done as previously described with primers annealing to the Thy1 promoter and the human APP transgene (*Lord et al., 2006*). The animals were housed under standard conditions with 12 hr dark–light cycles and unrestricted access to food and water. All animal procedures were in accordance with the National Institutes of Health Guide for the care and use of laboratory animals and approved by the Norwegian Food Safety Authority (project number: FOTS #11983).

## Viral transduction and delivery of fluorophores

Serotype 2/1 rAAV from plasmid construct pAAV-GFAP-GCaMP6f (*Chen et al., 2013*) was generated (rAAV titers about $1.0–6.0 \times 10^{12}$ viral genomes/ml) and used for visualizing astrocytic $Ca^{2+}$ signaling. GCaMP6f was amplified by PCR from pGP-CMV-GCaMP6f with 5′ *Bam*HI and 3′ *Hin*dIII, and subcloned into the recombinant rAAV vector pAAV-6P-SEWB (*Shevtsova et al., 2005*) for generating pAAV-*SYN*-GCaMP6f. The human *GFAP* promoter (*Hirrlinger et al., 2009*) was inserted with *Mlu*I and *Bam*HI into pAAV-*SYN*-GCaMP6f construct for obtaining pAAV-*GFAP*-GCaMP6f. Serotype 2/1 rAAVs from pAAV-*GFAP*-GCaMP6f were produced (*Tang et al., 2015*) and purified by AVB Sepharose affinity chromatography (*Smith et al., 2009*), following titration with real-time PCR (rAAV titers about $1.0–6.0 \times 10^{12}$ viral genomes/ml, TaqMan Assay, Applied Biosystems). For norepinephrine imaging experiments, mice were injected with ssAAV-9/2-*hSyn1*-GRAB-NE1m-WPRE-hGHp(A) (University of Zürich Viral Vector Facility). To visualize Aβ plaques, 7 mg/g methoxy-X04 (*Skaaraas et al., 2021*) dissolved in 0.1 M phosphate buffer saline (PBS) was injected intraperitoneally 24 hr prior to imaging. Also, WT littermates were injected with methoxy-X04 to rule out any confounding effects in one of the groups. We found that one injection provided enough signal to outline Aβ plaques for up to 3 days.

## Surgical preparation

Mice were anesthetized with isoflurane (3% for initiation, then 1–1.5% for maintenance) in room air enriched with 20% pure oxygen, and given buprenorphine 0.1 mg/kg s.c. preemptively for analgesia. Bupivacaine was administered subcutaneously over the skull and left for 10 min before a boat-shaped skin flap was removed. After removing the skin, a 2.5-mm-diameter craniotomy was drilled over the somatosensory cortex with center coordinates 3.5 mm lateral and –1.5 mm posterior to Bregma. Virus was injected (70 nl at 35 nl/min at 200 µm depth from the brain's surface) at three evenly spaced locations positioned to stay clear of large blood vessels, and a glass plug consisting of two coverslips glued together was placed in the craniotomy, slightly pressing the dura to prevent dural overgrowth (*Bojarskaite et al., 2020*). The surrounding area of the skull was sealed with cyanoacrylate glue and a layer of dental cement. Postoperatively the mice were given meloxicam 2 mg/kg for 2 days. Only animals with normal postoperative recovery were included in the study. Mice were left to recover for a minimum of 2 weeks before habituation to head-fixation and imaging.

## Two-photon microscopy

After recovery, the animals were imaged in layers 1–3 of the barrel cortex using a two-photon microscope (Ultima IV from Bruker/Prairie Technologies, Nikon 16 × 0.8 NA water-immersion objective model CFI75 LWD 16XW, Spectra-Physics InSight DeepSee laser, Peltier cooled photomultiplier tubes model 7422PA-40 by Hamamatsu Photonics K.K.). An excitation wavelength of 990 nm was used for GCaMP6f imaging, and separate recordings were made using 790 nm excitation light for imaging methoxy-X04, and 890 nm for Texas-Red labeled dextran for visualization of the vasculature. GRAB$_{NE}$ fluorescence was recorded at 920 nm excitation wavelength. Image time series were recorded at

30 Hz. 3D volume recordings of the morphology and plaque locations were performed in a subset of experiments. All images were recorded at 512 × 512 pixels with a resolution of approximately either 0.42 or 0.67 µm per pixel (FOV of 215 × 215 µm, or 343 × 343 µm, respectively). For imaging, the mice were head-fixed to a custom-built stage that allowed free locomotion on a wheel attached to a rotary encoder. During experiments, the mice were illuminated by an IR LED diode and monitored with an IR-sensitive camera. A second camera was used to capture the pupil dynamics (see below). Air puffs to elicit startle responses were delivered by a Picospritzer III (Parker). Instrument synchronization and data acquisition were performed with a custom-made LabVIEW 2015 (National Instruments) virtual instrument.

## Behavioral analysis

Mouse locomotion was recorded with a rotary encoder connected to the running wheel. Locomotion signal was captured using a National Instruments data acquisition card using a counter task in the NI Max software, activated through a custom LabVIEW (2015) VI. Data were processed with custom MATLAB scripts to classify running and quiet wakefulness. Criteria for run and quiet wakefulness episodes were validated by manual observation and defined as follows: running was defined as continuous segment of at least 4 s forward wheel motion at over 30°/s, no movement faster than 20°/s the last 10 s prior to start of running. Spontaneous running was not defined within 30 s following air puff. Quiet wakefulness was defined as continuous segment of at least 10 s duration with less than 2°/s locomotion, as well as no locomotor activity faster than 2°/s for 15 s before segment start. Quiet wakefulness episodes were not defined within 30 s following air puff. Quiet wakefulness periods were reviewed manually through the IR-surveillance video recordings to ensure animals were awake when sitting still.

## Pupillometry

Pupil size was recorded with a Basler Dart USB camera (daA1600-60um) with a 25 mm fixed focal length lens and 2× fixed focal length lens extender (Edmund optics, items #59–871 and #54–356). As two-photon imaging must be conducted in the dark where the pupil would be fully dilated, we illuminated the eye using 470 nm blue light fiber, with a shielding to avoid light contamination, to slightly constrict the pupil. Pupil size was delineated for time periods of isolated runs and startle using a custom-built tool developed in MATLAB 2020a. The ratio of pupil to eye-size ratio was calculated and used in the analysis.

## Image processing and analysis

Recordings were exclusively processed using MATLAB 2018a to 2020b. Imaging data were corrected for movement artifacts using the NoRMCOrre movement correction software (*Pnevmatikakis and Giovannucci, 2017*). We used our recently published imaging analysis toolbox 'Begonia' to remove motion artifacts, mark ROIs, and perform event-based $Ca^{2+}$ signal detection (ROA method). Astrocytic $Ca^{2+}$ signals were detected using the ROA method, and the ROA density (i.e., fraction [in %] of the compartment analyzed being active) and ROA frequency in the whole FOV or in anatomical subcompartments were calculated. Methods and algorithms related to this processing pipeline are described in detail in *Bjørnstad et al., 2021*. In brief, movement-corrected image time-series data of astrocytic $Ca^{2+}$ activity were preprocessed by variance stabilization (square root transform) and smoothing in the temporal and spatial domain to achieve a signal-to-noise ratio of at least 9. All image time series were for this reason smoothed with 14 frames in the temporal domain (~0.45 s) and two pixels in the x–y domain. Four times the median standard deviation of the variance stabilized data plus the baseline value for a pixel was then used to calculate a threshold for signal detection for each pixel. Once the time series had been binarized into signal and non-signal, adjoining x–y–t voxels (in time and space) were connected to ROAs. ROAs with a spatial extent of less than three pixels per frame and duration of less than three frames in time were not included in the analyses. For $GRAB_{NE}$ fluorescence, data were first downsampled from ~30 Hz to ~5 Hz and corrected for vascular artifacts (dips in fluorescence) due to increases in blood flow by software kindly shared by Prof. Kira Poskanzer at University of California San Francisco with colleagues and Prof. Guoqiang Yu with colleagues at Virginia Tech.

## 3D Aβ plaque mapping

Before any recording, z-stacks of images at 5 µm intervals taken at 512 × 512 pixels were recorded while illuminating the tissue with 790 nm laser light. The stacks started approximately at or below dura mater and extended 100–200 µm down. All other recordings were undertaken inside this mapped volume to ensure plaques outside the imaging plane were accounted for. For each imaged time series inside the volume, we recorded an additional single image at 790 nm to ensure the precise locations of plaques were known. Plaque locations were detected by binarizing the stack at a manually set threshold at which the morphology of plaques was visible. The binarized single images were used to manually align the 2D time-series data with the 3D plaque volume. The resulting 3D binary image had small points removed using the MATLAB function bwareaopen. Distances from plaque to nearest ROI (RP) were calculated using the 3D distance formula

$$RP = \sqrt{\left(x_{roi} - x_{plq}\right)^2 + \left(y_{roi} - y_{plq}\right)^2 + \left(z_{roi} - z_{plq}\right)^2}$$

## Tissue processing

All animals were sacrificed after final imaging procedures at the age of 18 months. The animals were anesthetized with Isofluran Baxter (IsoFlo, Abbot Laboratories) and intraperitoneally injected with ZRF mixture (zolazepam 3.8 mg/ml, tiletamine 3.8 mg/ml, xylazine 0.45 mg/ml, and fentanyl 2.6 µg/ml) before transcardial perfusion with 4°C 2% dextran in 0.1 M phosphate buffer (PB) for approximately 30 s, immediately followed by 4% formaldehyde (FA) in PB for 10 min at the speed of 6 ml/min. Following perfusion, the brains were extracted and post-fixed by immersion in the fixative at 4°C overnight protected from light. The tissue was stored in 0.1% FA in PB at 4°C protected from light until further processing. Cryoprotective steps in graded sucrose solution (10, 20, and 30% sucrose in PB) were performed before the brains were cut on a freeze microtome (Thermo Scientific Microm KS 34) in 40 µm free-floating coronary sections and stored in 0.1% FA in PB at 4°C protected from light until usage. In addition, eight tg-ArcSwe animals and eight WT littermates were sacrificed at 12 months of age and used for qPCR analysis. These animals were anesthetized as described above and decapitated. The brains were extracted, and the left hemisphere was dissected into the frontal cortex, hippocampus, cerebellum, and the rest of the brain. This tissue was frozen and stored in –80°C pending analysis.

## RNA isolation and real-time PCR

48 hr prior to RNA extraction, the samples were suspended in RNAlater-ICE (Ambion; Cat# AM7030). To isolate total RNA from the frontal cortex tissue samples, the RNeasy Mini Kit (QIAGEN, Hilden, Germany), including the on-column DNase digestion, was used. The RNA concentration and integrity were determined using a NanoDrop 2000c spectrophotometer (Thermo Fisher Scientific) and ethidium bromide visualization after agarose gel electrophoresis, respectively. Following the manufacturer's protocol, 1 µg of total RNA was reverse-transcribed into cDNA with Oligo (dT)$_{15}$ using the GoScript Reverse Transcription System (Promega, Madison, USA, Cat# A5001). All the cDNA samples were diluted in Tris-EDTA buffer (pH 8.0) to a final concentration of 2.5 ng/µl. Real-time PCR was carried out in a total volume of 20 µl, containing 2× AB Power SYBR Green PCR Master Mix (Thermo Fisher Scientific) with gene-specific primers (at a final concentration of 200 nM) and 2 µl cDNA samples. Amplification was performed on the StepOnePlus system (Applied Biosystems) with the following conditions: 95°C for 10 min, followed by 40 cycles of 95°C for 15 s and 60°C for 1 min, followed by melting curve analysis to check for unspecific products. Each sample was run in duplicates. Using the NormFinder software (*Andersen et al., 2004*), *HPRT1* was determined as an internal control for normalization of the gene expression. The primers were designed online using Primer BLAST and setting the amplicon size to a maximum of 200 bp. The primers designed span exon–exon junctions, and standards prepared as previously described (*Rao et al., 2021*). Details of the *Gfap* forward and reverse primer are presented in *Table 7*.

**Table 7.** Primer used for mRNA quantification.

| Gene | Protein name | Accession | Forward (5') | Reverse (3') |
|------|--------------|-----------|--------------|--------------|
| *Gfap* | Glial fibrillary acidic protein | NM_010277.3 | GCACTCAATACGAGGCAGTG | GCTCTAGGGACTCGTTCGTG |

## Immunohistochemistry

One section from each animal was chosen for quantification of astrogliosis and washed in PBS 0.01 M for 10 min, followed by two times in 0.1% TritonX100 in PBS (PBST) for 5 min. The PBST was removed and blocking (10% normal donkey serum [NDS], 1% bovine serum albumin [BSA], 0.5% Triton X100 in PBS) performed for 1 hr at room temperature. This was directly followed by incubation overnight at room temperature with primary antibodies (GFAP; host: mouse; diluted 1:1000; Sigma-Aldrich; Cat# MAB360. GFP; host: chicken; diluted 1:2000; Abcam; Cat# ab13970) diluted in antibody solution (ABS; 3% NDS, 1% BSA, 0.1% Triton X100 in PBS). The following day the sections were rinsed in 0.1% PBST two times for 1 min, followed by three times for 10 min. Secondary antibodies (CY5 donkey anti-mouse; Jackson ImmunoResearch Labs; Cat# 715-175-151; CY3 donkey anti-chicken; Jackson ImmunoResearch Labs; Cat# 703-165-155) were spun in a centrifuge for 10 min at 13,000 rpm, diluted 1:500 in ABS, and the sections incubated for 1 hr at room temperature. After the second incubation, the sections were washed in PBS for 10 min three times. Propidium iodide (diluted 1:5000 in 0.01 M PBS; Sigma-Aldrich; Cat# 04511 [Cellstain double staining kit]) for nuclear staining was added for 10 min, before rinsing the sections twice for 5 min in PBS. All sections were transferred to distilled water and mounted with ProLong Gold antifade reagent (Thermo Fisher Scientific; Cat# P36934). They were stored in –20°C protected from light until confocal imaging. For electron microscopy, EM standard procedure was followed for embedding and preparing of the tissue (*Yang et al., 2011*). For enhancing the contrast, uranyl acetate (Fluorochem) in double-distilled water and lead citrate was used. The sections were examined in a transmission electron microscope (TECNAI 12).

## Confocal imaging

GFP-positive astrocytes (indicating GCaMP6f expression) were used to locate the appropriate cortical area used for in vivo imaging. Only sections where we could locate positive GFP staining within the cortex corresponding to the image area were chosen for confocal imaging and further analysis. All single-plane and z-stack images were acquired using a Zeiss LSM 710 confocal microscope. To provide overview and verify that the correct area was identified, one tile scan of $3072 \times 3072$ pixels was achieved with a ×40 objective (1.20; water korr M27) using three channels (CY2, CY3, and CY5) for WTs and four channels (Dapi, CY2, CY3, and CY5) for tg-ArcSwe. Next, one z-stack of $2048 \times 2048$ pixels (×40 objective; 1.20; water korr M27) from within the GFP-positive area was acquired from cortical layer 2. All single-plane and z-stack images were obtained with identical settings. In addition, a z-stack of $1024 \times 1024$ pixels was obtained (×40 objective; 1.20; water korr M27) outside the located GFP-positive stained area. No postprocessing on the analyzed images was performed.

## 3D reconstruction analysis of astrocytes

This procedure was adapted from the detail outlined in *Tavares et al., 2017* that utilizes the free software plugin Simple Neurite Tracer (SNT) of Fiji ImageJ (*Longair et al., 2011*). Only astrocytes with a single nucleus where at least ⅔ of the circumference was covered by GFAP staining were selected for 3D reconstruction. Astrocytes with processes touching the borders of FOV were omitted. Eight randomly selected astrocytes from each z-stack (two from each image quadrant) obtained within the GFP-positive imaging area were used for analysis. To quantify and visualize the morphological complexity of the astrocytes, we analyzed the number of processes, total length of processes in µm, process thickness (µm³), and number of intersections (provided from the Sholl analysis).

## GFAP and GFP area fraction analyses

Z-projections based on the CY5 channel to visualize GFAP/GFP-positive astrocytes of all stack images were rendered using the Fiji ImageJ software (*Schindelin et al., 2012*). The images were blinded to the analyst, converted to 8-bit images and the scale removed. The threshold was manually adjusted so that only what was considered to be GFAP-positive staining was red, and a percentage value of positive staining in the image was obtained.

## Statistical analyses

Astrocytic $Ca^{2+}$ signals were studied by means of the ROA density – the fraction of the compartment with activity per time unit. Here, the area may be the entire FOV or we may limit ourselves to the area identified as belonging to cellular subcompartments; the astrocytic processes, somata, or endfeet. In

each trial, we had time series of ROA density lasting around 300 s (approximately 9000 frames). Within each trial, the startle period was defined as starting from the air puff at 150 s and lasting 600 frames (~20 s). In addition, one or more time periods within the trial could be identified as spontaneous runs or as periods of quiet wakefulness. A first important question concerns the choice of summary statistics adequately describing the $Ca^{2+}$ response in such periods of interest (runs and startle), which are dynamic behavioral states that entail both acceleration, steady locomotion and deceleration. We have studied the mean and max ROA density and the ROA density rise rate, which is defined as the *maximal increase in ROA density over a maximum of 50 frames*. Initial explorations indicated that the main results are fairly robust to the choice of window length, and 50 frames appeared to be a sensible choice compared to the kinetics of astrocytic $Ca^{2+}$ signals. The ROA density rise rate is meant to capture some of the dynamics in astrocytic $Ca^{2+}$ signaling and can be understood as the maximum acceleration inside the time period of interest. The rise rate has a high correlation with the maximum, and if the length of the window is increased sufficiently these two statistics tend to become almost identical (since most traces are close to zero at some point in the trial). See *Figure 4—figure supplement 3A*, which displays both the max ROA density and the rise rate in an example.

Pupil size measurements had a coarser time resolution than for the astrocytic $Ca^{2+}$ signals; see example in *Figure 4—figure supplement 3B*, showing the pupil sizes around a startle response. The pattern in the figure – a sharp increase in pupil size after air puff, before a gradual decline – was quite typical. Therefore, we chose to only consider the measurements in a time window of 6.67 s on each side of the air puff (or start of running). We defined the *pupil dilation* as the relative increase in the ratio of pupil diameter to eye diameter inside this window. In other words, we calculate the average ratio before the air puff (or start of running) and after the air puff, compute the difference, and divide by the average before the air puff.

For $GRAB_{NE}$ analyses, time series were segmented similar to the $Ca^{2+}$ imaging trials. Due to the slower response, change in $GRAB_{NE}$ fluorescence was calculated slightly different than the increases in $Ca^{2+}$ signaling as the difference between the median $GRAB_{NE}$ $\Delta F/F_0$ in the baseline period (10 s immediately prior to locomotion/startle) and during locomotion/starle. Pupil responses and locomotor responses were defined within the same time intervals for these trials.

## Interpreting hierarchical plots

We have chosen to present some of our data in the form of *hierarchical plots* (see *Figure 3B and C*, *Figure 4B and C*, *Figure 5B*, and figure supplements). These plots allow the reader to assess the degree of separation between the genotypes and at the same time get an impression of the *variation at different levels of the analysis* – in our case, the variation between different mice of the same genotype and the variation between repeated measurements on the same mouse. At the lowest level – the trial level – we have points representing the observations themselves. For the spontaneous runs, there are sometimes more than one run per trial and in that case the points are the median ROA density rise rate in these runs. The maximal number of spontaneous runs in a single trial was 5 (average: 1.6 runs per trial). At the middle level, we have the median ROA density rise rate for each mouse, and at the top level the median ROA density rise rate for each genotype. The lines between the levels indicate which observations belong to each mouse and to each genotype respectively. We have made similar plots for the max ROA density also. The hierarchical plots are a useful tool for exploratory analysis. They are also meant to promote transparency in scientific reporting and highlight the importance of intra-group variation. Still, it is important to realize that the impression conveyed by the plots might not be identical to the results from statistical modeling. In the plots, we do not include the influence of various technical and biological covariates that one typically would include in a statistical model. Some of the variation between observations belonging to the same mouse and between different mice of the same genotype may be explained by such covariates, as we will see in the next section.

## Modeling

Statistical analyses were conducted in R (version 4.1.1). The ROA density rise rate was modeled by linear mixed effect regression models, which were fitted using the glmmTMB package (*Brooks et al., 2017*). We conducted two sets of analyses: (1) to investigate potential differences in ROA density rise rate between the two genotypes and (2) to investigate the relationship between ROA density rise rate and the pupil dilation, including potential differences between the two genotypes with respect to this

relationship. For (1), the coefficient of primary interest is the one belonging to the genotype variable, while for (2) we are interested in the effect of pupil dilation on the ROA density rise rate, as well as the interaction between this pupil effect and the genotype. In both of these sets of analyses, we adjusted for the following fixed effect covariates: the level of magnification (µm per pixel) (two levels), the depth of the measurements (in µm), and the maximal speed in the relevant time window. We included random intercepts for each mouse (five WT and six tg-ArcSwe). We analyzed the ROA density max and mean values with similar models as the ones described here for the rise rate. For (1), the number of observations ranged between 77 and 117 trials with startle data (depending on the subcompartment) and between 72 and 109 episodes of spontaneous running, while for (2) the number of observations ranged between 60 and 86 for the startle data and between 35 and 44 for the spontaneous runs. There were less observations for the (2) analyses because some episodes/trials had missing or incomplete pupil measurements. Similar modeling was performed for GRAB$_{NE}$ analyses.

The sensitivity of our result to these modeling choices was assessed by various robustness checks (see next section). The adequacy of model assumptions was investigated by residual plots (*Hartig, 2022*). In the cases where the residual plots indicated deviations from the assumption of constant residual variance, we extended the model by allowing the residual variance to vary as a function of genotype. The reported p-values are based on the t-distribution, with degrees of freedom as provided from the glmmTMB package. No corrections for multiple comparisons were applied.

When analyzing the relationship between distance from the nearest plaque and astrocytic Ca$^{2+}$ signaling (*Figure 2E*), we considered the mean ROA density in each ROI (n = 10,988) in the quiet wakefulness episodes. If present, any effect of plaque proximity on Ca$^{2+}$ signaling should be discernible among ROIs observed in the same episode. The dashed lines in *Figure 2E* show the effect of distance on Ca$^{2+}$ signaling within each episode, and they form an uncertain picture: in some episodes, there is a weak positive relationship, with seemingly higher mean ROA density further away from plaques, while in many episodes there is a negative relationship, with somewhat higher mean ROA density close to plaques. The overall line is found by fitting a linear mixed effect model with mean ROA density as the response, with a fixed effect of distance and with each episode having its own random intercept and slope for the distance effect.

For the results presented in *Figure 1*, unless otherwise stated, the data are presented as mean ± standard error of the mean (SEM). A p-value equal to or less than 0.05 was considered statistically significant. Mann–Whitney *U*-test was used to analyze the number of processes, total length of processes in µm, process thickness in µm$^3$, and area fraction of positive GFAP staining. Two-way ANOVA followed by Sidak post-hoc comparison was used to analyze the number of intersections. Statistical analysis was performed in GraphPad Prism version 8.0.1 for Windows (GraphPad Software). For qPCR analysis, mean copy number per ng of total RNA was compared between genotypes by Mann–Whitney *U*-test in SPSS Statistics 26 (SPSS).

## Robustness checks

In order to check the robustness of the various statistical analyses, the stability of estimates and p-values was examined with respect to the length of window, for the rise rate response variable, and also sensitivity to individual mice and trials.

The type of sensitivity analysis performed is illustrated (see *Figure 2—figure supplement 2*) for the analysis of the uncoupling between pupil dilation and astrocytic Ca$^{2+}$ responses (see *Figures 3 and 4* for details about the data). Again, for the purpose of illustration, we focus on the interaction between pupil dilation and genotype for the ROA density rise rate response variable; see *Figure 2—figure supplement 2* for details. Similar analyses were performed for the main statistical models; none of these sensitivity analyses provided clear or strong evidence for any change in the main conclusions.

## Data and source code availability

A complete dataset of raw and processed data including two-photon microscopy traces, behavioral monitoring, and pupil tracking data is provided at https://doi.org/10.11582/2021.00100. Data analyses were performed with the Begonia toolkit (*Bjørnstad et al., 2021*), which is available at https://github.com/GliaLab/Begonia (copy archived at swh:1:rev:30447e22ca87fe7a2308b0029240bc4491a74518, *Bjørnstad, 2022*).

## Acknowledgements

This work was supported by the Medical Faculty, University of Oslo, the Olav Thon Foundation, the Letten Foundation, Norwegian Health Association, Alzheimerfondet - Civitan Foundation Norge, the Research Council of Norway (grants #249988, #302326, #271555/F20), and the South-Eastern Norway Regional Health Authority (grant #2016070). We acknowledge the support by UNINETT Sigma2 AS for making data storage available through NIRD, project NS9021K. Prof. Lars Lannfelt is gratefully acknowledged for help in the development of the tg-ArcSwe mouse model at Uppsala University. Prof. Kira Poskanzer at University of California San Francisco with colleagues and Prof. Guoqiang Yu with colleagues at Virginia Tech are gratefully acknowledged for sharing software for the analyses of GRAB$_{NE}$ fluorescence.

## Additional information

### Funding

| Funder | Grant reference number | Author |
| --- | --- | --- |
| Norges Forskningsråd | Grant 249988 | Rune Enger |
| Norges Forskningsråd | Grant 302326 | Rune Enger |
| Letten Foundation | Research support | Rune Enger |
| Olav Thon Stiftelsen | Olav Thon Award | Erlend A Nagelhus |
| Helse Sør-Øst RHF | Grant 2016070 | Rune Enger |
| Norges Forskningsråd | Medical Student Research Program | Kristin M Gullestad Binder |
| Helse Sør-Øst RHF | 2020039 | Rune Enger |
| Norges Forskningsråd | Grant 271555/F20 | Kristin M Gullestad Binder |

The funders had no role in study design, data collection and interpretation, or the decision to submit the work for publication.

### Author contributions

Knut Sindre Åbjørsbråten, Data curation, Formal analysis, Investigation, Methodology, Software, Validation, Visualization, Writing – review and editing; Gry HE Syverstad Skaaraas, Formal analysis, Investigation, Methodology, Validation, Visualization, Writing – review and editing; Céline Cunen, Formal analysis, Methodology, Validation, Visualization, Writing – review and editing; Daniel M Bjørn-stad, Data curation, Methodology, Software, Writing – review and editing; Kristin M Gullestad Binder, Formal analysis, Writing – review and editing; Laura Bojarskaite, Formal analysis, Investigation, Visualization, Writing – review and editing; Vidar Jensen, Methodology, Supervision, Writing – review and editing; Lars NG Nilsson, Resources, Writing – review and editing; Shreyas B Rao, Investigation, Writing – review and editing; Wannan Tang, Methodology, Resources, Writing – review and editing; Gudmund Horn Hermansen, Formal analysis, Investigation, Validation, Visualization, Writing – review and editing; Erlend A Nagelhus, Funding acquisition, Project administration, Supervision, Professor Nagelhus died tragically in January 2020. Up until then he contributed significantly to the project; Ole Petter Ottersen, Conceptualization, Supervision, Writing – original draft, Writing – review and editing; Reidun Torp, Conceptualization, Investigation, Methodology, Project administration, Supervision, Writing – review and editing; Rune Enger, Conceptualization, Formal analysis, Funding acquisition, Investigation, Methodology, Project administration, Software, Supervision, Visualization, Writing – original draft, Writing – review and editing

### Author ORCIDs

Rune Enger (iD) http://orcid.org/0000-0001-9418-7117

### Ethics

The study was performed in strict accordance with the Guide for the Care and Use of Laboratory Animals of the National Institutes of Health and approved by the Norwegian Food Safety Authority (project number: FOTS #11983).

### Decision letter and Author response

Decision letter https://doi.org/10.7554/eLife.75055.sa1
Author response https://doi.org/10.7554/eLife.75055.sa2

---

## Additional files

### Supplementary files

• Transparent reporting form

### Data availability

The numerical data for the statistical analyses in Figures 3-5 are available as Source data 1. The complete dataset is available at https://doi.org/10.11582/2021.00100.

The following dataset was generated:

| Author(s) | Year | Dataset title | Dataset URL | Database and Identifier |
| --- | --- | --- | --- | --- |
| Åbjørsbråten KS, Cunen C, Bjørnstad DM, Binder KMG, Jensen V, Nilsson LNG, Rao SB, Tang W, Hermansen GHH, Nagelhus EA, Ottersen OP, Torp R, Enger R, Ghes S | 2021 | Impaired astrocytic Ca2+ signaling in awake Alzheimer's disease transgenic mice | https://doi.org/10.11582/2021.00100 | NIRD Research Data Archive, 10.11582/2021.00100 |

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
