## [Editor Report]

This article is of broad interest to readers in the astrocyte and Alzheimer's disease (AD) fields, and it utilizes state-of-the-art techniques to simultaneously record astrocyte calcium and animal behavior. The work provides new insight into astrocyte calcium responses in AD, which has important implications for astrocyte pathophysiology. Overall, the data are of high quality and well analyzed.

---

## [Decision Letter]

**Decision letter after peer review:**

Thank you for submitting your article "Impaired astrocytic ca^2+^ signaling in awake Alzheimer's disease transgenic mice" for consideration by *eLife*. Your article has been reviewed by 3 peer reviewers, including Mark T Nelson as Reviewing Editor and Reviewer #1, and the evaluation has been overseen by Richard Aldrich as the Senior Editor. The following individual involved in review of your submission has agreed to reveal their identity: Jillian L Stobart (Reviewer #2).

Essential revisions:

1) The heavy reliance on pupil response, which is only partially linked to NE, takes the manuscript into a tautological reasoning. Authors should consider using another read-out of NA activity. For instance they have the expertise and technical capability necessary to employ GRAB-NE sensors. In the final analysis, too much of the weight of this study rests on pupillary dilation measurements that are just not reproducible enough to bear the burden; this is especially true for pupil responses induced by spontaneous locomotion.

2) The authors should conduct some experiments in anesthetized Tg-ArcSwe mice to hone in on the idea that anesthesia is the causal factor in discrepancies observed in the field, and to strengthen the take-home message of the study, experiments could be done in a paired fashion (i.e. comparison of un-anesthetized and under isoflurane anesthesia, in the same mice), rather than as a separate cohort of mice. These experiments could be provided later. Regardless, the authors should greatly soften their conclusions about anesthesia without firm evidence.

3) Why were tg-ArcSwe mice chosen over other AD models?

4) Figure 2A: In the leftmost panel, does the sudden increase in ROA signal density at ~60 s correspond to initiation of a spontaneous running event? (Not specified in the text or legend.) What does the relatively higher density of signals centered around ~25 s reflect-background activity? How do differences in these x-y-t plots between WT and Tg-ArcSwe mice relate to phenotypic differences described? Are these data used in calculation of ROA density rise rate? If so, how? Description of methods on this point (imaging processing and analysis) is not sufficient to even get the gist of how this is done (over-relies on a previous publication).

5) Figure 2B: Rate (top panels) and density (bottom panels) are presumably normalized relative values. If so, this should be indicated, and the normalization procedure used should be described. Is this presentation style a standard in the field or a home-grown method? It's certainly not intuitively relatable and only becomes comprehensible after prolonged staring. The authors should be a little less stingy with text that might help readers orient.

6) Figure 2C: What are the statistics on these regression analyses? Visually, WT results appear to be uncorrelated (approximately vertical "eyeball" regression line), centered around a relative increase of ~0.2, whereas tg-ArcSwe results look like shotgun blasts with a hint of a negative correlation. Hardly compelling.

7) Figure 3C: Again, what are the statistics on these analyses? Although trends in pupillary responses are more evident with the startle paradigm, the data are too messy to form the basis of a firm conclusion.

8) The authors make the reasonable argument that a means for measuring astrocytic ca^2+^ signaling in the absence of anesthesia is necessary, but the "compared-to-what" aspect needed to support their argument is missing. The case for quantitative and/or qualitative differences in astrocytic ca^2+^ signals in the absence of anesthesia would be considerably strengthened by including a comparison of startle response-astrocytic ca^2+^ relationships (or other locomotion-independent relationships) between anesthetized and unanesthetized animals.

9) The reactive gliosis assessment has a tacked-on quality; seems more like something that would be included in a basic characterization of the tg-ArcSwe model. Its inclusion might make sense if the authors had in mind teasing out a relationship between reactive gliosis and astrocytic ca^2+^ signaling.

10) Clearly lay out the goals of the study, providing sufficient description of all the various aspects-role of NA signaling in AD, NA induction of astrocytic ca^2+^ signals, behavior-induced (NA-dependent) ca^2+^ signals, novel technique for real-time monitoring of astrocytic ca^2+^ signals, etc-for the reader to follow along.

11) Include a cartoon showing how all the many moving parts of the study are related.

12) Provide a better description of the imaging processing and analysis procedures (including event-based detection aspect) in Methods so that the reader doesn't have to go to a previous publication to understand the basic workflow.

13) Include better (or at least some) description of what panels illustrate in figure legends (see comments in Public Review).

14) Provide statistics on regression analyses; a supplemental figure might be helpful in this regard.

15) The authors did not provide a justification for why they chose this mouse model and to use 15-month-old animals. Further information about the symptoms and the known disease progression of this strain would be beneficial. Previous studies used AD models that contained mutations in PSEN1 (e.g. APPPS1 mice). Also, these studies used 5-9-month-old animals. This alone could account for differences in results, particularly if the stage of disease is not comparable.

16) Authors study was performed in 15-months old mice when the plaque load is significant. I am wondering if they have done or would consider doing some recordings in younger animals from the same line, in which the plaque burden is still absent (or significantly less) and age-matched controls. This could help determine whether the decoupling they observed is due to the plaque load itself or whether it preexists to the plaques. Considering that sleep is among the first alterations observed in AD and considering the growing body of evidence implicating astrocytes (and NE) in sleep, this could yield some very interesting and important results.

17) It is unclear to me why authors picked this specific mouse model of AD. I don't feel very strongly about any AD model in particular but I wonder if authors could provide a rational/justification for using this particular model.

18) It is unclear to me whether Methoxy-X04 was injected in AD mice in all experiments and, more importantly, whether it was injected in control animals as well? If it was only injected in AD mice, I do think authors should conduct some control experiments where they confirm the absence of effect of this probe alone on astrocytes ca^2+^ signaling and morphology, especially in the case of AD astrocytes since they engulf/phagocytose plaques and might thus accumulate large amounts of Methoxy-X04. This might have already been performed by others but I am not aware of it.

19) For the conclusion of Figure 2 to be fully valid, authors would need to show quantitative data indicating that the full dynamic range of functional pupil dilation in AD mice is statistically identical to controls (which seems to be the case from Figure 3 panel C). Otherwise, an alternative interpretation would simply be an overall attenuation of the input-output curve of the pupil dilation neurological reflex.

20) The only major concern I have with this study, and I am genuinely surprised authors do not discuss it, is that GFAP is used to drive GCaMP expression and, as authors elegantly show, the GFAP promoter is strongly overactive in AD astrocytes by nearly 10-folds. This presumably drives a 10-fold stronger expression of GCaMP in AD mouse astrocytes. Since GCaMP acts as a strong ca^2+^ chelator/buffer, this alone could disrupt ca^2+^ signaling, baseline ca^2+^-dependent mechanisms in astrocytes, and/or fail to report ca^2+^ activity in a way that is comparable to controls, in such a way that this could partially explain the impaired ca^2+^ signaling observed in AD mice. Would it be possible for authors to quantify GCaMP protein expression accurately (i.e. not by proxy of baseline fluorescence) and rule this out? Alternatively, would it be possible to provide control data indicating that the expression level of GCaMP in AD mouse astrocytes is not deleterious? I think this is a crucially important control experiment.

21) Lines 209-212, authors present data that is not statistically significant as a de facto difference between control and AD mouse ca^2+^ signaling, on the basis that the p-value is close to 0.05. Whether it is biologically relevant or not, the "0.05 p-value" is an international statistical standard and authors should stick to it. I agree that there is a "trend towards" (line 209-210): is it possible that the sample size for the study was underpowered? If not, then the biological difference is not statistically significant and authors should remove statements of "trend".

---

## [Author Response]

Essential revisions:1) The heavy reliance on pupil response, which is only partially linked to NE, takes the manuscript into a tautological reasoning. Authors should consider using another read-out of NA activity. For instance they have the expertise and technical capability necessary to employ GRAB-NE sensors. In the final analysis, too much of the weight of this study rests on pupillary dilation measurements that are just not reproducible enough to bear the burden; this is especially true for pupil responses induced by spontaneous locomotion.

We agree with the reviewers that using the GRAB_NE_ sensor in our current experimental setup could provide valuable mechanistic insight. We have therefore run an additional round of imaging experiments where we subject tg-ArcSwe mice and littermate controls expressing GRAB_NE_ to the same experimental protocol as for the ca^2+^ imaging. In both spontaneous running and startle responses we find attenuated GRAB_NE_ fluorescence increases in tg-ArcSwe mice compared to WT littermates, potentially – at least partially – explaining the differences in astrocytic ca^2+^ responses in AD mice. More specifically, the effect of pupil dilation on GRAB_NE_ fluorescence is lower in tg-ArcSwe mice, suggesting that pupil responses are not so tightly coupled to cortical norepinephrine release in the transgenic mice.

One new figure (Figure 5) presenting the GRAB_NE_ data with a corresponding Results section (“Attenuated noradrenergic signaling in tg-ArcSwe mice”) and 2 tables (tables 5 and 6) have been added. We have further expanded our discussion about the new results as well as limitations of the study.

2) The authors should conduct some experiments in anesthetized Tg-ArcSwe mice to hone in on the idea that anesthesia is the causal factor in discrepancies observed in the field, and to strengthen the take-home message of the study, experiments could be done in a paired fashion (i.e. comparison of un-anesthetized and under isoflurane anesthesia, in the same mice), rather than as a separate cohort of mice. These experiments could be provided later. Regardless, the authors should greatly soften their conclusions about anesthesia without firm evidence.

The aim of the current study was to analyze signaling in awake, behaving animals, and we see the literature as so compelling when it comes to the detrimental effect of anesthesia on astrocytic ca^2+^ signaling that we found it of little relevance to recapitulate studies made under anesthesia. Anesthesia clearly removes behaviorally and neuromodulatory driven astrocytic ca^2+^ signaling (by at least 90%) (Thrane et al., 2012; Ding et al., 2013). This also fits well with our own experimental studies both from anesthetized and un-anesthetized mice (Enger et al., 2017; Enger et al., 2015; Heuser et al., 2018; Bojarskaite et al., 2020). Accordingly, our ca^2+^ event frequency is orders of magnitude more frequent than what was presented in Delekate et al., 2014 (even though a direct comparison is not entirely fair as we have used a different method for quantification of ca^2+^ signals). Our experimental model also differs quite considerably with previous studies, both in terms of age of inclusion, AD model, ca^2+^ sensors used and experimental setup for a fair comparison to be made.

Thus we did not aim to confirm or disprove the well-documented astrocytic hyperactivity evident in previous studies in anesthetized mice. Importantly, we do find clearly pathological signaling, although rare compared to the behaviorally driven ca^2+^ signaling in the tg-ArcSwe mice, that we believe represents what has been previously reported. That being said, we agree that a direct comparison of anesthetized or awake tg-ArcSwe mice and WT littermates could provide some relevant information about pathological ca^2+^ hyperactivity in AD. Even though anesthesia prominently dampens physiological astrocytic ca^2+^ signaling, this may not be the case with the pathological “hyperactive” ca^2+^ signals in AD. It is important to note that we consider the relatively straightforward experimental design suggested by the reviewer for determining the effects of anesthesia to be a bit simplistic. Notably, a “cross-over design” is not compatible with artificial ventilation and control of blood gasses, which we deem necessary for maintaining physiological parameters in anesthetized mice, as self-breathing mice would typically (to some degree) be hypoxic and hypercapnic if breathing room air, or hyperoxic and hypercapnic if oxygen is supplied. Moreover, isoflurane is probably an agent that would be particularly effective in quenching astrocytic ca^2+^ signaling (Thrane et al., 2012). With the relatively extensive effort it would take for a proper investigation of this topic, we rather focussed our resources on developing sophisticated behavioral paradigms and analytical methods than revisiting the effects of anesthesia on astrocytic ca^2+^ signaling.

Finally, at the current time we do not have enough mice at the appropriate age group to both perform GRAB_NE_ experiments and experiments in anesthesia, and prioritized the experiments we believed had the biggest potential to add to the story. To meet the critique by the reviewers, we have now softened our conclusions about the effects of anesthesia except in cases where we consider the statement clearly justified by studies in WT mice.

3) Why were tg-ArcSwe mice chosen over other AD models?

The tg-ArcSwe model is an AD model that expresses amyloid plaques in the parenchyma throughout most of the brain, including cortex, but also expresses strong vascular amyloid deposits (as also is the case in AD patients) (Lillehaug et al., 2014, Neurobiology of Aging). The model expresses plaques that are biochemically close to the composition of plaques in human AD (Philipson et al., 2009). Moreover, we have extensive experience with the mouse model from previous studies (Syverstad Skaaraas et al., 2021; Lillehaug et al., 2014; Yang et al., 2017), and the plaque load and distribution, and behavioral phenotype is well characterized (Codita et al., 2010; Lillehaug et al., 2014). Most importantly, however, we deem the tg-ArcSwe AD model particularly relevant for studying astrocytic physiology, due to the prominent vascular deposits and the close anatomical relationship between astrocytic endfeet and blood vessels and distinct consequences to astrocytic endfoot expression of key proteins like aquaporin-4. Two sentences have been added to the introduction.

4) Figure 2A: In the leftmost panel, does the sudden increase in ROA signal density at ~60 s correspond to initiation of a spontaneous running event? (Not specified in the text or legend.) What does the relatively higher density of signals centered around ~25 s reflect-background activity? How do differences in these x-y-t plots between WT and Tg-ArcSwe mice relate to phenotypic differences described? Are these data used in calculation of ROA density rise rate? If so, how? Description of methods on this point (imaging processing and analysis) is not sufficient to even get the gist of how this is done (over-relies on a previous publication).

Indeed both these increases in ROAs represent ca^2+^ activity coupled to locomotor activity, but the locomotor activity at ~25 s was not large enough to be defined as a run. We added labeling in the figure and a few sentences to Figure 2A legend to explain these activation patterns.

The *x-y-t* plots shown in figure 2 are representative examples of the data used for the ROA analyses. The data used for all analyses are essentially the data used to generate these *x-y-t* plots of ROAs (binary masks of signal (ROA) vs background). A trace of mean ROA activity per frame (i.e. fraction of area of signal per frame) has been used to calculate the maximum ROA density and ROA density rise rate (and mean values as presented in Figure 3 —figure supplement 1). The ROA method has now been better explained in Results in a new paragraph pertaining to statistical modeling, as well as in the methods section.

5) Figure 2B: Rate (top panels) and density (bottom panels) are presumably normalized relative values. If so, this should be indicated, and the normalization procedure used should be described. Is this presentation style a standard in the field or a home-grown method? It's certainly not intuitively relatable and only becomes comprehensible after prolonged staring. The authors should be a little less stingy with text that might help readers orient.

These hierarchical plots are not a standard in the field, and therefore we had added a section in Materials and methods about interpretation of hierarchical plots. Also, to make this information more easily available, the figure legends have been expanded with more explanations. We believe these plots are particularly well suited for displaying hierarchical data, as it displays variability in several levels of the hierarchy, as opposed to box-and-whisker plots, violin plots or similar.

Values as indicated in the x-axis of these graphs represents ROA density rise rate and max ROA density without any further normalization, as ROA density is expressed as a fraction of the analyzed area (FOV, astrocyte soma, process, endfoot). We have now changed the fractional number to percent as this may be more intuitive for the reader. We have also added labeling for the x-axes in the hierarchical plots, and more text in the figure legends.

6) Figure 2C: What are the statistics on these regression analyses? Visually, WT results appear to be uncorrelated (approximately vertical "eyeball" regression line), centered around a relative increase of ~0.2, whereas tg-ArcSwe results look like shotgun blasts with a hint of a negative correlation. Hardly compelling.

See point 7.

7) Figure 3C: Again, what are the statistics on these analyses? Although trends in pupillary responses are more evident with the startle paradigm, the data are too messy to form the basis of a firm conclusion.

Answer to point 6 and 7: (See also point 1.) We agree that there is a relatively large spread of data points in these scatter plots. However, the slopes and p-values are clearly different between the two genotypes, and the effects are robust to various statistical controls (see Figure 2 —figure supplement 2). The statistics to analyze these relationships have been provided in the section “Two-photon imaging of awake-behaving tg-ArcSwe mice“ in the Results section. It is important to note that the statistical model includes more covariates than what is shown in the scatter plot (the dots merely represent median activation per trial). Moreover, 6 tables are added with results from the statistical models used to generate all p-values for comparisons.

8) The authors make the reasonable argument that a means for measuring astrocytic ca^2+^ signaling in the absence of anesthesia is necessary, but the "compared-to-what" aspect needed to support their argument is missing. The case for quantitative and/or qualitative differences in astrocytic ca^2+^ signals in the absence of anesthesia would be considerably strengthened by including a comparison of startle response-astrocytic ca^2+^ relationships (or other locomotion-independent relationships) between anesthetized and unanesthetized animals.

See a more complete answer to point 2 that is essentially raising the same concern. Again, the rationale of the study was for the first time to investigate a different mode of astrocytic ca^2+^ signaling than what had been reported previously, namely signaling associated with behavior. We consider comparing startle responses in anesthetized vs. awake mice would be highly influenced by the particular anesthesia employed (depth, type of anesthetic agent) and maybe of lesser value.

We agree that a comparison between anesthetized and awake mice could provide important information, but do not have mice of the appropriate age to perform these experiments at the current time, and the work needed for a proper analysis of these questions is extensive as it would require several different types of anesthesia, anesthesia depths and probably also a new way of inducing “startle” that would work in anesthesia (foot shock or electrical stimulation of of the whisker pad, which are methods that we do not have established in our laboratory).

As detailed in point 2, we have softened our conclusion where we believe appropriate.

9) The reactive gliosis assessment has a tacked-on quality; seems more like something that would be included in a basic characterization of the tg-ArcSwe model. Its inclusion might make sense if the authors had in mind teasing out a relationship between reactive gliosis and astrocytic ca^2+^ signaling.

We did initially plan to include levels of GFAP as a covariate in our statistical modeling. However, the GFAP levels came out as non-significant in the statistical modeling, and there was no clear relationship between GFAP level and ca^2+^ signaling in the AD mice. We believe the data still are of interest to include in the story as gliosis is not well described in this particular mouse model before in the neocortex, and also may provide one possible explanation why pupil responses are not so clearly correlated to astrocytic ca^2+^ signaling anymore, as reactive astrocytes may express a different repertoire of receptors and signaling pathways.

10) Clearly lay out the goals of the study, providing sufficient description of all the various aspects-role of NA signaling in AD, NA induction of astrocytic ca^2+^ signals, behavior-induced (NA-dependent) ca^2+^ signals, novel technique for real-time monitoring of astrocytic ca^2+^ signals, etc-for the reader to follow along.

We agree that the background of NE in relation to both AD and astrocytic ca^2+^ signals should be presented more comprehensively. We have now expanded/reorganized the Introduction to better justify the rationale for the study.

11) Include a cartoon showing how all the many moving parts of the study are related.

We have now re-organized figure 1 to include a cartoon illustrating the experimental protocol.

12) Provide a better description of the imaging processing and analysis procedures (including event-based detection aspect) in Methods so that the reader doesn't have to go to a previous publication to understand the basic workflow.

A better description has now been provided in Methods, subsection “Image processing and analysis”.

13) Include better (or at least some) description of what panels illustrate in figure legends (see comments in Public Review).

Figure legends have been expanded throughout to better explain the panels.

14) Provide statistics on regression analyses; a supplemental figure might be helpful in this regard.

We have expanded the Results section and added 6 tables with the results from statistical modeling.

15) The authors did not provide a justification for why they chose this mouse model and to use 15-month-old animals. Further information about the symptoms and the known disease progression of this strain would be beneficial. Previous studies used AD models that contained mutations in PSEN1 (e.g. APPPS1 mice). Also, these studies used 5-9-month-old animals. This alone could account for differences in results, particularly if the stage of disease is not comparable.

See point 16.

16) Authors study was performed in 15-months old mice when the plaque load is significant. I am wondering if they have done or would consider doing some recordings in younger animals from the same line, in which the plaque burden is still absent (or significantly less) and age-matched controls. This could help determine whether the decoupling they observed is due to the plaque load itself or whether it preexists to the plaques. Considering that sleep is among the first alterations observed in AD and considering the growing body of evidence implicating astrocytes (and NE) in sleep, this could yield some very interesting and important results.

For both 15 and 16: For justification of why Arc-Swe model was used, please see point 3. We intended here to study astrocytic ca^2+^ signaling in a disease state where the mice would express a clear phenotype, both in terms of cognitive function, and pathomorphological changes, such as a clear plaque load, hoping to probe whether relevant differences of astrocytic ca^2+^ signals were present in AD that potentially could contribute to perturbed cognition. We have added more information about the symptoms and disease progression of this mouse strain in the Introduction and in the Materials and methods. We agree that experiments in younger mice are worth pursuing for future experiments, focussing on the earlier states in the disease progression and this is now commented upon in the Discussion.

17) It is unclear to me why authors picked this specific mouse model of AD. I don't feel very strongly about any AD model in particular but I wonder if authors could provide a rational/justification for using this particular model.

See also point 3. Information about our motivation to use this particular model is now added in the Introduction.

18) It is unclear to me whether Methoxy-X04 was injected in AD mice in all experiments and, more importantly, whether it was injected in control animals as well? If it was only injected in AD mice, I do think authors should conduct some control experiments where they confirm the absence of effect of this probe alone on astrocytes ca^2+^ signaling and morphology, especially in the case of AD astrocytes since they engulf/phagocytose plaques and might thus accumulate large amounts of Methoxy-X04. This might have already been performed by others but I am not aware of it.

All mice, tg-ArcSwe and controls, were injected with Methoxy-X04 to avoid any systematic confounding effects in only one group. This information is now added in the Results section and Materials and methods.

19) For the conclusion of Figure 2 to be fully valid, authors would need to show quantitative data indicating that the full dynamic range of functional pupil dilation in AD mice is statistically identical to controls (which seems to be the case from Figure 3 panel C). Otherwise, an alternative interpretation would simply be an overall attenuation of the input-output curve of the pupil dilation neurological reflex.

We have analyzed the full recordings of pupil data and find pupil dynamic response ranges are similar in both groups. A plot with the comparison is added to Figure 4 —figure supplement 2. No statistically significant differences were present between the groups (p = 0.93).

20) The only major concern I have with this study, and I am genuinely surprised authors do not discuss it, is that GFAP is used to drive GCaMP expression and, as authors elegantly show, the GFAP promoter is strongly overactive in AD astrocytes by nearly 10-folds. This presumably drives a 10-fold stronger expression of GCaMP in AD mouse astrocytes. Since GCaMP acts as a strong ca^2+^ chelator/buffer, this alone could disrupt ca^2+^ signaling, baseline ca^2+^-dependent mechanisms in astrocytes, and/or fail to report ca^2+^ activity in a way that is comparable to controls, in such a way that this could partially explain the impaired ca^2+^ signaling observed in AD mice. Would it be possible for authors to quantify GCaMP protein expression accurately (i.e. not by proxy of baseline fluorescence) and rule this out? Alternatively, would it be possible to provide control data indicating that the expression level of GCaMP in AD mouse astrocytes is not deleterious? I think this is a crucially important control experiment.

We did assess GFP expression by confocal microscopy of sections from the FOVs used for 2P imaging and found comparable levels between the groups (median fluorescence: 28 ± 10.3 a.u. in AD vs. 25 ± 10.9 a.u. in WT, p = 0.88, area fraction of GFP labeling: 66.1 ± 4.44 in AD vs. 70.5 ± 5.42 in WT). Even if quantification of fluorescence from brain slices is an imprecise measurement, it would clearly detect large differences in sensor expression as suggested by the reviewer. Why we do not see a significantly increased GCaMP expression in the AD mice with much more GFAP expression is not entirely clear, but could be due to rAAV infectious rates in normal vs. reactive astrocytes for instance. Clearly these data are important to report, and are now to be found in Figure 2 —figure supplement 3. Moreover, we find a similar level of ca^2+^ signaling in quiet wakefulness in tg-ArcSwe compared to WT mice, suggesting a comparable level of sensor expression between the two groups, as basal astrocytic ca^2+^ levels are known to determine also ca^2+^ signal frequency (see “Local Resting ca^2+^ Controls the Scale of Astroglial ca^2+^ Signals”, King et al., Cell Reports 2020). A sentence has been added in the discussion pertaining to the point raised by the reviewer.

21) Lines 209-212, authors present data that is not statistically significant as a de facto difference between control and AD mouse ca^2+^ signaling, on the basis that the p-value is close to 0.05. Whether it is biologically relevant or not, the "0.05 p-value" is an international statistical standard and authors should stick to it. I agree that there is a "trend towards" (line 209-210): is it possible that the sample size for the study was underpowered? If not, then the biological difference is not statistically significant and authors should remove statements of "trend".

We have amended the manuscript in accordance with the reviewer’s comments.